# Insights from year-long measurements of air-water $CH_4$ and $CO_2$ exchange in a coastal environment

Mingxi Yang*, Thomas G. Bell, Ian J. Brown, James R. Fishwick, Vassilis Kitidis, Philip D. Nightingale, Andrew P. Rees, Timothy J. Smyth

Plymouth Marine Laboratory, Prospect Place, Plymouth, UK PL1 3DH.
* Correspondence to M. Yang (miya@pml.ac.uk)

**Abstract**. Air-water $CH_4$ and $CO_2$ fluxes were directly measured using the eddy covariance technique at the Penlee Point Atmospheric Observatory on the southwest coast of the United Kingdom from September 2015 to August 2016. The high frequency, year-long measurements provide unprecedented detail into the variability of these Greenhouse Gas fluxes from seasonal to diurnal and to semi-diurnal (tidal) timescales. Depending on the wind sector, fluxes measured at this site are indicative of air-water exchange in coastal seas as well as in an outer estuary. For the open water sector when winds were off the Atlantic Ocean, $CH_4$ flux was almost always positive (annual mean of ~0.05 mmol m$^{-2}$ d$^{-1}$) except in December and January, when $CH_4$ flux was near zero. At times of high rainfall and river flow rate, $CH_4$ emission from the estuarine-influenced Plymouth Sound sector was several times higher than emission from the open water sector. The implied $CH_4$ saturation (derived from the measured fluxes and a wind speed dependent gas transfer velocity parameterization) of over 1000% in the Plymouth Sound is within range of in situ dissolved $CH_4$ measurements near the mouth of the river Tamar. $CO_2$ flux from the open water sector was generally from sea-to-air in autumn and winter and from air-to-sea in late spring and summer, with an annual mean flux of near zero. A diurnal signal in $CO_2$ flux and implied partial pressure of $CO_2$ in water ($pCO_2$) are clearly observed for the Plymouth Sound sector and also evident for the open water sector during biologically productive periods. These observations suggest that coastal $CO_2$ efflux may be underestimated if sampling strategies are limited to daytime only. Combining the flux data with seawater $pCO_2$ measurements made in situ within the flux footprint allows us to estimate the $CO_2$ transfer velocity. The gas transfer velocity vs. wind speed relationship at this coastal location agrees reasonably well with previous open water parameterizations in the mean, but demonstrates considerable variability. We discuss the influences of biological productivity, bottom-driven turbulence and rainfall on coastal air-water gas exchange.

## 1 Introduction

Methane ($CH_4$) and carbon dioxide ($CO_2$) are two of the most important Greenhouse Gases (GHGs). Their tropospheric abundances have increased over the last few hundred years primarily due to human activities, with the fastest increases in the last

50 years (Hartmann et al. 2013). Highly dynamic estuarine and coastal regions can be important sources and sinks of these GHGs. Understanding the emissions and uptake of these gases by coastal waters and how they change is directly relevant to the fulfillment of the United Nations Framework Convention on Climate Change (UNFCCC) Paris 2016 agreement. We argue in

this paper that the eddy covariance (EC) technique, with a temporal resolution of tens of minutes to hours, is an excellent method for long-term monitoring of coastal air-sea $CH_4$ and $CO_2$ fluxes.

There has been much debate over the causes of the recent tropospheric $CH_4$ trend, from varying wetland (e.g. Pison et al. 2013; Schaefer et al. 2016; Nisbet et al. 2016) and fossil fuel (e.g. Helmig et al. 2016; Rice et al. 2016) emissions, to changes in the atmospheric oxidative capacity (e.g. Rigby et al. 2017). Inland aquatic systems may be important sources of tropospheric

$CH_4$ (e.g. Borges et al., 2015). Similarly, due to benthic methanogenesis, large surface $CH_4$ supersaturations of thousands of percent have been observed in estuaries (e.g. Upstill-Goddard et al., 2000; Middelburg et al., 2002). $CH_4$ concentrations in estuaries can be influenced by processes including biological productivity, organic carbon input, benthic and particle-derived $CH_4$ production, oxygen content, as well as the hydrodynamics (e.g. Upstill-Goddard et al. 2000; 2016). In regions of intense benthic methanogenesis, gas bubbles supersaturated with $CH_4$ episodically rise through the water column to the surface (e.g.

Dimitrov, 2002; Kitidis et al., 2007). This process of ebullition will result in $CH_4$ emissions that are not quantified using air-sea flux calculations based on seawater $CH_4$ concentration (see below). In coastal seas, $CH_4$ saturation tends to be lower than in estuaries, but is still much greater than 100% (e.g. mean >200% for European shelf waters; Bange et al. 2006). Consequently, estuaries and coastal seas tend to have much greater $CH_4$ emissions per unit area than the open ocean (Bange et al. 2006; Forster et al. 2009).

Seawater $CO_2$ levels are primarily determined by solubility (temperature-dependent) and the balance between primary production and respiration by the biological community. Seasonal and geographical differences in seawater temperature and biological activity mean that the surface ocean can act as a net source or sink of $CO_2$, depending on location and time of the year (Khatiwala et al. 2013; Houghton 2003). Models estimate that $2.4\pm0.5$ GtC $yr^{-1}$ of $CO_2$ (a quarter of anthropogenic emissions) have been absorbed by the global ocean over the last decade (Le Quéré et al. 2018). Shelf seas, despite their relatively small

area, support high primary productivity, cause a large drawdown of $CO_2$ in the mean (Frankignoulle and Borges, 2001; Chen et al. 2013), and might be responsible for as much as 10-40% of global oceanic carbon sequestration (Muller-Karger et al. 2005; Cai et al. 2006; Chen et al. 2009; Laruelle et al. 2010). Estuaries, on the other hand, are generally net sources of $CO_2$ for the atmosphere (e.g. Frankignoulle et al. 1998). Inner estuaries are estimated to emit about 0.3 GtC $yr^{-1}$ of $CO_2$ globally (Laruelle et al. 2010; Cai 2011). Most of this $CO_2$ emission is due to the degradation of allochthonous organic matter rather than a direct

input of dissolved inorganic carbon (Borges et al. 2006). The direction of net air-sea $CO_2$ flux is less certain in coastal areas that are influenced by riverine outflow and anthropogenic activities (Chen et al. 2013). Kitidis et al. (2012) showed a gradient of

increasing air-to-sea $CO_2$ flux with distance offshore in the western English Channel. The coastal seas may have been

heterotrophic during preindustrial conditions and thus a net source of $CO_2$ due to organic carbon degradation (e.g. Smith and

Hollibaugh, 1993). Some studies (e.g. Andersson and Mackenzie, 2004; Cai, 2011) predict that shallow seas will become a net

sink (or a reduced source) of $CO_2$ in the future due to rising atmospheric $CO_2$ levels and increased inorganic nutrient inputs.

Modeling of the carbonate chemistry and hence $CO_2$ flux in the North Western European shelf is hindered partly because of the

uncertain representation of riverine influence (Artioli et al. 2012).

To quantify the impacts of estuarine and coastal emissions on the atmospheric $CH_4$ and $CO_2$ burden, an indirect method

requiring the inventories of air-sea concentration difference ($\Delta C$) and the gas transfer velocity ($K$) is usually utilized: Flux =

$K \cdot \Delta C$. Coastal areas tend to be highly dynamic, with greater spatial and temporal variability in physics and biology than the

open ocean. This heterogeneity poses serious challenges to observational and modeling efforts aimed at constraining coastal air-

sea GHG fluxes. Dissolved gas concentrations may be affected by tides, currents, mixed layer processes, and benthic/pelagic

interactions. The sheltered nature of the coastal seas, coupled with freshwater input, often result in stratification (e.g. Sims et al.

2017), where biological processes can more quickly modify the near-surface dissolved gas concentrations. Mixed layer

dynamics can vary on a diurnal timescale, due for example to buoyancy forcing (e.g. Esters et al. 2018). The atmospheric

concentrations of GHGs at coastal locations also vary as a function of wind direction, airmass history, and boundary layer

processes (e.g. Yang et al. 2016a). Estuaries and coastal seas in mid-latitudes also tend to experience large seasonal variability,

which affect the dissolved gas concentrations (e.g. Crosswell et al. 2012; Joesoef et al. 2015).

The transfer velocity ($K$) primarily depends on near surface turbulence, and over the ocean is generally parameterized as

a function of wind speed (e.g. Wanninkhof et al. 2009). Currents and resultant bottom-driven turbulence significantly affect gas

exchange in shallower waters, resulting in $K$ values that can be much higher than predicted based on wind speed alone

(O'Connor and Dobbins, 1958; Borges et al. 2004; Ho et al. 2014). Rainfall is highly episodic, but may be important for gas

exchange because it generates additional turbulence and/or alters near-surface gas concentrations (e.g. Ho et al. 1997; Zappa et

al. 2009; Turk et al. 2010). Variability in biogeochemical processes could also affect $K$ by changing the surface tension and

modifying the turbulence at the air-sea interface. Pereira et al. (2016) observed a gradient of increased sea surface surfactant

activity from the open sea towards the coast, which reduced the gas transfer velocity by approximately a factor of two in their

laboratory tank simulations. Thus a wind speed-only dependent representation of $K$ is probably even less appropriate for coastal

environments than for the open ocean.

Measuring the fluxes directly with the eddy covariance technique is an ideal way to study the many controlling factors

of air-sea exchange in dynamic and heterogeneous environments such as shallow waters and coastal seas. It also allows us to test

the appropriateness of the indirect flux calculations. Furthermore, compared to shipboard EC observations, measuring fluxes

from a stationary tower has the advantage of not requiring any motion correction on the winds (see Edson et al. 1998). This means that flux and $K$ measurements at a coastal location are potentially more accurate, especially at high wind speeds when the motion correction for a moving platform would become large. Only a few coastal stations exist worldwide that have reported

air-sea $CO_2$ fluxes by EC on a seasonal timescale, such as Östergarnsholm station in the Baltic Sea (Rutgersson et al. 2008), the Utö Atmospheric and Marine Research Station also in the Baltic Sea (Honkanen et al. 2018), Punta Morro in Baja California, Mexico (Gutíerrez-Loza and Ocampo-Torres, 2016), and Qikirtaarjuk Island in the Canadian Arctic (Butterworth and Else, 2018). In the case of the Östergarnsholm station, concurrent measurements of partial pressure of seawater $CO_2$ (p$CO_2$) from a nearby buoy allow for the determination of the $CO_2$ gas transfer velocity. $CH_4$ sensors with sufficient measurement frequency

and precision for the EC methods have only been developed in recent years (Yang et al. 2016b). We are not aware of any published long-term air-sea $CH_4$ fluxes by the EC method.

In this paper, we describe a year-long set of air-water $CO_2$ and $CH_4$ flux measurements by EC at the coastal Penlee Point Atmospheric Observatory. The high frequency fluxes allow us to characterize their variability across a range of time scales (semi-diurnal to diurnal to seasonal). Combining these data with in situ observations of dissolved gas concentrations as well as

supporting physical and biogeochemical measurements enables us to quantify the gas transfer velocity at this coastal location and examine its controls.

**2 Experimental**

The Penlee Point Atmospheric Observatory (PPAO, 50° 19.08' N, 4° 11.35' W;

http://www.westernchannelobservatory.org.uk/penlee/) was established in May 2014 on the southwest coast of the United Kingdom. Understanding the controls of coastal air-sea exchange is one of the main scientific foci at this site. Yang et al (2016a, 2016b) demonstrated that the PPAO is a suitable location to measure air-sea exchange by the EC method.

**2.1 Eddy covariance fluxes**

Atmospheric $CH_4$ and $CO_2$ mixing ratios were measured at a frequency of 10 Hz using a Los Gatos Research (LGR) Fast Greenhouse Gas Analyzer (FGGA, enhanced performance model) between September 2015 and August 2016. As described in detail by Yang et al (2016a), two Gill sonic anemometers (Windmaster Pro and R3) are installed on a mast on the rooftop of PPAO (~18 m above mean sea level). For this paper, wind data from the Windmaster Pro sonic anemometer were used between September 2015 and March 2016. Since March 2016, wind data from the R3 sonic anemometer (not operational for the first 6

months of this annual study) were preferred because of its higher precision and better performance during heavy rain events. The effect of rain on the EC gas flux measurement is discussed in the Supplementary Materials.

The gas inlet tip, located ~30 cm below the Windmaster Pro sonic anemometer centre volume, is connected to the LGR via ~18 m long PFA tubing (3/8'' outer diameter). First a scroll pump (BOC Edwards XDS-35i) until 16 October 2015 and then a rotary vane pump (Gast 1023) were used to pull sample air through the inlet tubing, an aerosol filter (2 μm pore size, Swagelok SS-6F-05), and the LGR. The aerosol filter became laden with sea salt over time and the filter elements were replaced approximately every ~2 months. As a result, the volumetric flow rate through the LGR varied between 23 and 78 LPM, which affected the lag time and the high frequency attenuation of the fluxes. The lag time was determined from a maximum lag correlation analysis between $CO_2$ and the instantaneous vertical wind velocity ($w$), varying from about 2.7 to 9.0 s. The strong atmosphere-biosphere flux of $CO_2$ when winds were from land aided our determination of this lag time. The high frequency flux attenuation was estimated from the instrument response time (see Yang et al. 2013, 2016a) and a wind speed dependent correction was applied to the flux data (representing a ~15% gain in the mean).

Fluxes of $CH_4$ and $CO_2$ were initially computed in 10-minute intervals from the covariance of their lag-shifted dry mixing ratios and $w$. Wind velocities were streamline corrected using the standard double rotation method (Tanner and Thurtell, 1969) on a 10-minute basis. Evaluations of the EC momentum transfer against the expected rate (Figure S2 in Supplementary Materials), as well as stationarity in winds and gas mixing ratios, are used to quality control the 10-minute flux data. The filtered 10-minute fluxes are further averaged to hourly and also six-hourly intervals to reduce random noise. See Yang et al. (2016a, 2016b) for further details on data processing, quality control, and measurements of momentum and sensible heat fluxes. Horizontal wind speed measurements are corrected for flow distortion and adjusted to a neutral atmosphere at 10-m height (see Supplementary Materials).

## 2.2 Flux footprints

The theoretical flux footprint model of Kljun et al. (2004) predicts the upwind distance of maximum flux contribution ($X_{max}$) and the distance of 90% cumulative flux contribution ($X_{90}$). The semi-diurnal tidal range at this location is large (up to 6 m during spring tide), effectively raising the EC measurement height above water at low tide and reducing it at high tide. For a neutral atmosphere, the Kljun et al (2004) model estimates $X_{max}$ and $X_{90}$ to be approximately 0.4 and 1.1 km at the highest tide and 0.6 and 1.6 km at the lowest tide. As described in more detail by Yang et al. (2016a), stable and unstable atmospheres are predicted to increase and decrease $X_{max}$ as well as $X_{90}$ by a few tens of percent, respectively.

In this paper we focus on air-water transfer over two different wind sectors. When winds are from the southwest (180-240°), the eddy covariance flux footprint is over open water with a depth of approximately 20 m at $X_{max}$. When winds are from the northeast (45-80°), the footprint is over the fetch-limited Plymouth Sound (approximately 5-6 km wide), which is ~10 m

deep and more influenced by the outflow of the Tamar estuary (Siddorn et al. 2003; Uncles et al. 2015). See Supplementary Materials Figure S1 for a map of the site and the approximate flux footprints.

**2.3 Seawater measurements**

We used the Plymouth Marine Laboratory's Research Vessels (RV) *Quest* and *Plymouth Explorer* to study the spatial heterogeneity in this coastal environment. Underway seawater measurements on the *Quest* from ~3 m depth include $pCO_2$ (Kitidis et al. 2012), salinity, temperature, chlorophyll, and dissolved oxygen. As a part of the Western Channel Observatory sampling program (http://www.westernchannelobservatory.org.uk), the *Quest* made approximately weekly trips to the L4 station (50° 15.0' N, 4° 13.0' W; ~6 km south of PPAO) and fortnightly trips to the E1 station (50° 02.6' N, 4° 22.5' W; ~20 km south

of PPAO). These visits were always during the daytime. On the way back to Plymouth from L4 and E1, the *Quest* often idled at about 600 m to the south/southwest of PPAO for approximately 10 minutes, enabling the collection of underway measurements within the open water flux footprint of PPAO. The ship also passed through the Plymouth Sound flux footprint of PPAO en route back into port.

Seawater samples were taken at the L4 station from a CTD rosette. For $CH_4$ analysis, discrete seawater samples were

collected directly into 500ml borosilicate bottles from Niskin bottles using clean Tygon tubing. Sample bottles were overfilled by three times their volume to eliminate air bubbles, poisoned with 100μl of a saturated mercuric chloride solution and returned to the laboratory where they were transferred to a water bath at 25°C and temperature equilibrated for a minimum of one hour before analysis. Samples were analysed for $CH_4$ by single-phase equilibration gas chromatography using a flame ionisation detector similar to that described by Upstill-Goddard et al. (1996). Samples were typically analysed at PML within 2 weeks of

collection and calibrated against three certified (±5%) reference standards (Air Products Ltd), which are traceable to NOAA WMO-N2O-X2006A.

The other PML vessel (a hard-bottomed RHIB, *Plymouth Explorer*) was used to occasionally sample the estuary Tamar from the upper freshwater section near Gunnislake to the lower saltwater section near the Plymouth Sound in 2017 and 2018. This is a part of the NERC-funded LOCATE (Land Ocean Carbon Transfer; http://www.locate.ac.uk) research programme.

Discrete seawater samples were collected at stations from the near surface into 500ml borosilicate bottles with care taken to eliminate air bubbles. Analysis for dissolved $CH_4$ was performed as described above.

**3 Results and discussion**

Over the one year of measurements, variability in physical parameters was large: wind speed at times exceeding 20 m s⁻

¹, and seawater temperature varying between about 7 and 18°C. Chlorophyll a concentration ranged between about 0.2 and 5 mg

m$^{-3}$, with generally higher values from late spring to early autumn than in winter. Time series of ancillary data (meteorological parameters, Tamar river flow, and surface ocean physical and biogeochemical parameters) are shown in the Supplementary Materials (Figures S3 to S6). This region can be roughly characterised by a windier, wetter autumn and winter, and a calmer, dryer spring and summer. Southwesterly winds off the Atlantic Ocean (annual mean wind speed of ~8 m s$^{-1}$) occurred more frequently in the winter months, resulting in higher precipitation rates and greater riverine discharge. During these conditions, the temperatures in the sea surface and air were similar throughout the entire year, resulting in fairly small air-sea temperature differences and modest sensible heat fluxes (monthly average of typically -20 to 20 W m$^{-2}$). As a result, the atmosphere was often close to neutral stability with a monthly mean Monin-Obukhov stability parameter ($z/L$) between -1.7 and 0.04.

### 3.1 CH$_4$ fluxes and implied seawater concentrations

Figure 1 shows the air-sea flux of CH$_4$ over the one-year measurement period. Flux data gaps are due to either wind direction outside of air-water sectors or instrumental failure. As shown by Yang et al. (2016b), under ideal conditions (moderate winds and steady atmospheric mixing ratio) the random uncertainty in the LGR CH$_4$ flux due to band-limited instrumental noise is on the order of 0.02 and 0.01 mmol m$^{-2}$ d$^{-1}$ for one-hour average and six-hour average, respectively. In comparison, the standard deviation ($\sigma$) in the six-hour averaged CH$_4$ flux for the open water sector is about 0.05 mmol m$^{-2}$ d$^{-1}$ (computed over the entire year). Thus much of the rapid temporal fluctuations in the measured CH$_4$ flux appear to be driven by natural variability (due to changes in watermass within the flux footprint, wind, etc.), rather than due to random instrumental noise. CH$_4$ flux from the open water sector at times shows semi-diurnal (tidal) variability (consistent with Yang et al. 2016a). We note that most of what appear to be negative CH$_4$ fluxes are within the uncertainty of the EC measurement, and are not significantly different from zero.

To more clearly illustrate the seasonal variability, the means and 25/75 percentiles of the six-hour averaged CH$_4$ fluxes are computed in monthly intervals (Figure 2). CH$_4$ flux was consistently positive, indicating emission of CH$_4$ from these coastal waters. The only exception was during the months of December and January, when CH$_4$ flux was near zero. The annual mean CH$_4$ flux from the open water sector was 0.047 (standard error, or SE of 0.008) mmol m$^{-2}$ d$^{-1}$ when computed from monthly mean fluxes and 0.039 (SE of 0.003) mmol m$^{-2}$ d$^{-1}$ when directly computed from six-hour mean fluxes. Wind directions that enable air-sea flux measurements did not occur with the same frequency throughout the year. For example, southwesterly winds were less frequent in spring (30% of time in March-May 2016) than in winter (60% of time in January 2016). Thus annual averages computed directly from the six-hour fluxes are more heavily weighted by the periods with a high proportion of valid flux measurements. In contrast, annual averages computed from the monthly means give more equal weight to all the months.

The annual mean $CH_4$ flux here is largely consistent with previous coastal estimates (e.g. Upstill-Goddard et al. 2016) and roughly an order of magnitude greater than estimates of $CH_4$ flux for the open ocean (e.g. Forster et al. 2009).

CH4 flux from the Plymouth Sound sector was noticeably higher than flux from the open water sector, with an annual mean of about 0.108 (SE of 0.026) mmol $m^{-2}$ $d^{-1}$. This enhancement in the flux was particularly noticeable at times of high rainfall and river discharge rate, with fluxes over 0.2 mmol $m^{-2}$ $d^{-1}$ in February 2016. During the dry summer months of 2016

(May and June), $CH_4$ fluxes from the two wind sectors were comparable. Northerly winds occurred only 7.4% of the time overall during the one-year study period. Thus the seasonal variability in $CH_4$ emission from the Plymouth Sound is less well represented than emission from the open water sector.

We briefly compare our measured fluxes with existing estimates of riverine $CH_4$ emission. The 1-$km^2$ resolution UK National Atmospheric Emissions Inventory (NAEI, http://naei.defra.gov.uk) reports a natural $CH_4$ emission source of 0.17 mmol

$m^{-2}$ $d^{-1}$ averaged over the area of the Plymouth Sound for year 2013. Our annual mean flux from the Plymouth Sound wind sector is about 64% of the NAEI estimate. Based on in situ measurements of dissolved $CH_4$ concentrations in six major UK estuaries, Upstill-Goddard et al. (2016) estimated $CH_4$ emissions of 4.3 Gg $yr^{-1}$ for UK outer estuaries (using a total outer estuarine area of 1894 $km^2$). If we crudely assume that the Plymouth Sound is a representative outer UK estuary, scaling up our mean flux from this wind sector to the total outer estuarine area of 1894 $km^2$ yields an annual flux of 1.2 Gg $yr^{-1}$. This is lower

than the estimate from Upstill-Goddard et al. (2016), likely because according to their survey the $CH_4$ saturation from the Tamar is fairly low compared to some of the other major UK estuaries. The UK has a 12429 km long coastline. If the PPAO open water footprint is representative of the nearest 1.4 km (i.e. typical $X_{90}$ of our fluxes, see Section 2.2) of the UK coast, our measurements crudely extrapolate to a total $CH_4$ flux of 4.8 Gg $yr^{-1}$ for the UK coastal seas. This order-of-magnitude estimate is made from a mean flux of 0.047 mmol $m^{-2}$ $d^{-1}$ and a total coastal sea area of 12429 km by 1.4 km. We are not able to use PPAO

EC flux data to provide estimates for $CH_4$ emission from the inner estuary, where fluxes are likely higher per unit area (Upstill-Goddard et al. 2016).

We wish to disentangle the processes that control the gas transfer velocity ($K$) from the processes that control the air-water concentration difference ($\Delta C$). We first compute the implied seawater $CH_4$ and $CO_2$ concentrations from the eddy covariance fluxes by assuming a parameterization of the gas transfer velocity. Here the sea-minus-air concentration difference

($\Delta C$) is computed by dividing the EC flux by the wind speed-dependent $K$ from Nightingale et al. (2000) (adjusted for ambient Schmidt number by the exponent of -0.5). Adding the atmospheric concentration to $\Delta C$ yields the implied seawater concentration. At low wind speeds, both the flux and $K$ trend towards zero. To avoid excessive noise from dividing one small number by another, implied seawater concentrations at wind speeds lower than 5 m $s^{-1}$ are discarded. Note that we apply the

Nightingale et al. (2000) wind speed-based $K$ parameterization here largely because it is commonly used and lies between the very strong and the very weak wind speed dependent relationships.

Implied seawater $CH_4$ concentration from the open water flux footprint ranges from about 3 to 26 nM on a monthly interval (mean of 14 nM; see Figure 3). It is often convenient to represent dissolved $CH_4$ as a saturation level relative to the atmosphere (saturation = $CH_{4w}$ / ($CH_{4a} \bullet sol_{CH4}$) $\bullet 100$, where $CH_{4w}$ and $CH_{4a}$ are waterside and airside concentrations; $sol_{CH4}$ is the $CH_4$ solubility from Wanninkhof 2014). $CH_4$ saturations are shown in Figure S8. The lowest implied $CH_4$ concentration occurred in winter and corresponded to a saturation level close to 100%. The highest implied $CH_4$ concentration was from April to November, with an average saturation level of about 600%. The temperature and salinity-dependent solubility of $CH_4$ varies by only ~14% from summer to winter at this location. The seasonal variability in $CH_4$ concentration and saturation is thus more due to changing biological processes (methanogenesis and/or $CH_4$ oxidation) and hydrodynamics than due to dissolution (i.e. seasonal temperature changes). For the Plymouth Sound flux footprint, the implied concentration ranges from 9 to 99 nM (mean of 37 nM, corresponding to about 1200% saturation), with the highest values in late winter and early spring. These implied concentrations and saturations are compared with nearby dissolved $CH_4$ measurements in Section 3.3.3. We note that any contribution to $CH_4$ emission from ebullition would have been included in the EC flux measurements, potentially resulting in higher implied seawater $CH_4$ concentrations than the measured dissolved concentrations.

Based on measurements from April to June 2015 at PPAO, Yang et al. (2016a) observed that $CH_4$ flux from the open water flux footprint varied with the timing of the tide, but not with the tide height. Specifically, $CH_4$ flux tended to be the highest during the first ~4 hours after low water. This was attributed to the outflow of a lower salinity surface layer from the Tamar river during rising tide around the Penlee headland. A subtle semi-diurnal variability in $CH_4$ flux can be seen in Figure 6B, where adjacent six-hour mean fluxes always alternated between higher and lower values during these few days. The same general tidal pattern is apparent over an annual cycle in the implied saturation level of $CH_4$. On average, the implied $CH_4$ saturation within the open water sector was about 40% higher during rising tide than during falling tide.

### 3.2 CO$_2$ fluxes and implied seawater concentrations

Figure 4 shows the air-sea flux of $CO_2$ over the one-year measurement period. The random uncertainty in the LGR $CO_2$ flux, estimated from the band-limited instrumental noise, is on the order of 4 and 2 mmol m$^{-2}$ d$^{-1}$ for one-hour average and six-hour average measurements, respectively (Yang et al. 2016b). The standard deviation in the six-hour averaged $CO_2$ flux for the open water sector is about 20 mmol m$^{-2}$ d$^{-1}$ (computed over the entire year), substantially greater than the random uncertainty due to instrumental noise. The rapid temporal fluctuations in $CO_2$ flux are likely to be driven by variability in winds as well as variability in seawater pCO$_2$. The latter is unlikely to be fully captured by weekly or monthly seawater sampling.

The means and 25/75 percentiles of the six-hour averaged $CO_2$ fluxes are computed in monthly intervals (Figure 5).

$CO_2$ flux from the open water sector was generally from sea-to-air in autumn and winter (up to 37 mmol m$^{-2}$ d$^{-1}$) and from air-to-sea in spring and early-summer (as much as -26 mmol m$^{-2}$ d$^{-1}$). The seasonality in $CO_2$ flux is consistent with seawater $pCO_2$ observations by Litt et al. (2010) and Kitidis et al. (2012) from the same region, and is partly driven by biology. Figure S7 shows that in situ $pCO_2$ generally decreased with increasing chlorophyll a concentrations during this annual study. $CO_2$ flux from the Plymouth Sound sector appeared to be more positive than from the open water sector in some months.

A three-day time series of $CO_2$ flux from July 2016 is shown in Figure 6A. Winds were consistently from the southwest during this period, varying from about 3 to 12 m s$^{-1}$ (Figure 6C). $CO_2$ flux during this period was clearly different between day (mean of about -13 mmol m$^{-2}$ d$^{-1}$) and night (mean of about +9 mmol m$^{-2}$ d$^{-1}$), with an overall mean of about -2 mmol m$^{-2}$ d$^{-1}$. Daytime $pCO_2$ measurements on the *Quest* from the 7$^{th}$ and 12$^{th}$ of July imply a $\Delta pCO_2$ of about -40 μatm and a net flux into the water. The EC flux is consistent in sign with $\Delta pCO_2$ during the day, but not at night. The computed transfer velocity of $CO_2$

($K_{CO2,660}$, see Section 3.4) using the linearly interpolated daytime $pCO_2$ measurements yielded positive values during the day (as expected), but negative values at night (which is not physically possible). The positive $CO_2$ flux at night is unlikely to be caused by a nocturnal flux footprint that overlaps with land because both sensible heat and $CH_4$ fluxes are consistent with air-sea exchange. The air temperature was about 1.2 °C warmer than the water temperature, implying a slightly stable atmosphere and a flux footprint that extends a few tens of percent further upwind from the PPAO site than in a neutral atmosphere (Kljun et al.

2004). The measured sensible heat flux averaged -9 W m$^{-2}$ and showed little diurnal variability. Similarly, $CH_4$ flux was positive (sea-to-air) and did not vary with the time of day.

The most likely reason for the negative nighttime $K_{CO2,660}$ is that seawater $pCO_2$ varied diurnally, probably due to a combination of biological and dynamical processes. Wind speed was generally higher at night during these few days and the measured fluxes imply that the actual $\Delta pCO_2$ (see next sections for this calculation) changed from about -40 μatm during the day

to about 15 μatm at night. Similar diurnal cycles (with slightly reduced magnitudes) have been observed in $pCO_2$ measurements in the western English Channel by Marrec et al. (2014) and Litt et al. (2010). We note that a daytime CTD cast on 12$^{th}$ July 2016 showed a mixed layer at the L4 station of only ~10 m depth. Entrainment of deeper water could contribute towards a higher surface $pCO_2$ at night. A diurnal cycle in $CO_2$ flux was not obvious during times of expected evasion (sea-to-air flux). These periods of positive $CO_2$ flux occurred in autumn and winter when biological productivity was low and the water column was

mixed to the bottom.

The annual mean $CO_2$ flux was 3.9 (SE of 4.9) mmol m$^{-2}$ d$^{-1}$ when computed from monthly mean fluxes (Figure 5) and 1.3 (SE of 1.3) mmol m$^{-2}$ d$^{-1}$ when directly computed from six-hour mean fluxes. If we sub-sample our EC observations to the period of 10:00–16:00 UTC only, the annual mean $CO_2$ flux becomes 2.5 (SE of 4.9) mmol m$^{-2}$ d$^{-1}$ when computed from

monthly mean fluxes and -1.0 (SE of 2.2) mmol m$^{-2}$ d$^{-1}$ when directly computed from six-hour mean fluxes. These results

highlight the value of continuous flux measurements and suggest that $CO_2$ flux estimates based only on daytime $pCO_2$

measurements may be biased towards greater seawater net uptake for coastal environments such as the western English Channel.

Monthly averaged implied seawater $CO_2$ concentrations from the two flux footprints are shown in Figure 7. The

greatest supersaturation in $CO_2$ is observed in late autumn in the open water sector, with values exceeding 500 µatm. The

greatest undersaturation in $CO_2$ is observed in late spring and early summer, coinciding with an increase in chlorophyll a

concentration at the nearby L4 station (Figure S6). Average implied $pCO_2$ within the Plymouth Sound is 32 µatm higher than

$pCO_2$ within the open water flux footprint during months when fluxes were available for both wind sectors. This difference

between the outer estuary and the coastal seas qualitatively agree with previous observations of supersaturated $pCO_2$ in the river

Tamar (Frankignoulle et al. 1998).

Average implied seawater $CO_2$ saturation for the open water sector over the entire year is about 100% in the daytime

and slightly higher at night (Figure 8). In contrast, a marked diurnal variability in $CO_2$ saturation is observed for the Plymouth

Sound sector, with a higher saturation level at night than during the day. Compared to the open water sector, Plymouth Sound is

more sheltered and influenced by the Tamar outflow, thus subject to greater near surface stratification and possibly different

biological processes. The diurnal variability we observed is important in the context of estuarine $CO_2$ (and carbonate system)

observations that are only carried out during daytime. Our findings suggest that such a daytime-only monitoring strategy may

underestimate estuarine $pCO_2$ and by extension the efflux of $CO_2$ to the atmosphere.

Semi-diurnal variability as a result of the tide is not obvious in the $CO_2$ flux or the implied $CO_2$ saturation. This

suggests that the influence of the Tamar estuary on $pCO_2$ within the PPAO flux footprints is limited, consistent with the in situ

$pCO_2$ measurements (see Section 3.3.2). The diurnal variability in $pCO_2$ might also be confounding any semi-diurnal tidal

signal.

It is worth noting that our implied seawater GHG concentrations would be overestimated if the in situ gas transfer

velocity were higher than the wind speed dependent parameterization of Nightingale et al. (2000). For example, bottom-driven

turbulence could enhance the gas transfer velocity (e.g. Borges et al. 2004; Ho et al. 2014). We discuss the effects of depth and

current velocity on gas exchange in Section 4.3.

**3.3 Spatial homogeneity of the study region**

The estimation of the gas transfer velocity $K$ requires concurrent measurements of flux and seawater concentration

within the flux footprint. Seawater $pCO_2$ was typically measured once or twice a week, and only some of the measurements

were made within the PPAO flux footprints. Observations of dissolved $CH_4$ were even scarcer and unfortunately none of them

were made within the flux footprints. In order to relate the high frequency EC fluxes to the discrete in situ dissolved gas

concentrations, we first evaluate the spatial homogeneity of our study region using shipboard seawater measurements.

### 3.3.1 Variability in salinity

Previous modeling studies (Siddorn et al. 2003; Uncles et al. 2015) show that freshwater discharge from the Tamar

estuary mainly flows along the western edge of Plymouth Sound and bends around PPAO towards the southwest. In Figure 9,

we compare underway salinity measured within the PPAO flux footprints (open water to the southwest as well as the Plymouth

Sound to the northeast) with near-coincidental *Quest* observations at the L4 station (6 km south of PPAO). Compared to the L4

station, mean salinity was 1.2 % and 2.1% lower in the open water and Plymouth Sound footprints, respectively. Periods of low

salinity both within the footprints and at L4 coincided with the greatest outflow from the Tamar estuary. These observations

indicate that the Tamar outflow influences this entire region; unsurprisingly water is generally fresher within the Plymouth

Sound than in the open water flux footprint. We next assess how much this riverine outflow affects the seawater $CO_2$ and $CH_4$

concentrations within the flux footprints of PPAO and thus the measured fluxes.

### 3.3.2 Variability in seawater $pCO_2$

The underway in situ $pCO_2$ measured within the PPAO flux footprints is compared with near-coincidental observations

from the *Quest* at the L4 station in Figure 10. The highest $pCO_2$ measured both within the footprints and at L4 occurred at times

of large riverine discharge. This is seemingly consistent with a Tamar influence (e.g. Frankignoulle et al. 1998) but may also be

driven by the seasonality in $pCO_2$ (Kitidis et al. 2012). As shown in Figures S9-S11 of the Supplementary Materials, fast

responding sea surface temperature and chlorophyll a were not noticeably different between the flux footprints and L4, while

dissolved oxygen was slightly lower within the footprints. $pCO_2$ measurements within both flux footprints were very similar to

$pCO_2$ at the L4 station. The apparent agreement for $pCO_2$ could be in part because the measurement with a 'shower head'

equilibrator has an integration time of 8 minutes (Kitidis et al., 2012). The *Quest* usually only idled for ~10 minutes within the

open water flux footprint and did not idle within the Plymouth Sound footprint. It is possible that the $pCO_2$ spatial variability is

under-represented in the $pCO_2$ measurements due to the fairly slow response time of the equilibrator.

In situ $pCO_2$ measurements from the Plymouth Sound footprint and from the open water footprint (plus L4, since they

are not distinguishable) are shown in Figure 7, along with the 100% saturation value with respect to the atmosphere. Implied

$pCO_2$ for the open water sector and the in situ $pCO_2$ within the open water footprint (plus L4) broadly agree. Constraining the

implied $pCO_2$ estimate to during the day further improves the agreement with the in situ $pCO_2$ measurements (also daytime only,

see Section 3.2). These observations suggest that the direct impact of the Tamar outflow on $pCO_2$ in the open water flux footprint at PPAO is fairly small relative to the air-sea concentration difference as well as other sources of variability.


### 3.3.3 Variability in dissolved $CH_4$

Dissolved $CH_4$ was not measured within either of the PPAO flux footprints. Here we look at how our implied $CH_4$ concentrations from the fluxes compare to measurements of dissolved $CH_4$ in the river Tamar and at L4. On four separate days in April 2017, July 2017, January 2018, and April 2018, the *Plymouth Explorer* was used to sample dissolved $CH_4$ from the

upper reaches of the Tamar to the seaward end during a falling tide . $CH_4$ in the estuarine part of the Tamar in general correlated inversely with salinity (Figure 11). For example, in April 2017 the $CH_4$ concentration was 491 nM at a salinity of 4.7 (upper Tamar), 274 nM at a salinity of 29.3 (lower Tamar), 15 nM at a salinity of 34.2 (at the mouth of the Tamar in the Plymouth Sound), and 2.4 nM at a salinity of 35.2 (L4). These correspond to $CH_4$ saturation values of ~10000% at a salinity of 29.3 and ~600% at a salinity of about 34.2 during this transect. The highest $CH_4$ concentration was measured in July 2017 following

heavy rainfall, while relatively low $CH_4$ were observed in January and April 2018. The measurements from the river Tamar in 2001 by Upstill-Goddard et al. (2016) are within the range of these more recent transects. Long-term observations of surface dissolved $CH_4$ at L4 between October 2013 and July 2017 indicate a mean (±σ) saturation of $123 \pm 60\%$.

The implied seawater $CH_4$ concentrations for the Plymouth Sound sector (Section 3.1) are within range of the in situ measurements in the lower Tamar and near the Plymouth Sound. In contrast, implied seawater $CH_4$ concentrations for the open

water sector are on average about four times higher than the in situ measurements at L4. Thus, while $pCO_2$ within the open water flux footprint of the PPAO agrees reasonably well with $pCO_2$ at L4, this is very likely not the case for $CH_4$. The differences in salinity (Figure 9) and in dissolved oxygen (Figure S11) indicate that the water masses within the open water flux footprint and at L4 are not identical.

Two features of the $CH_4$ concentration vs. salinity relationship are particularly relevant for the interpretation of our $CH_4$

flux measurements. First, the variability in dissolved $CH_4$ concentration in the Tamar is very large. For example, $CH_4$ concentration at a salinity of about 30 varies by a factor of 40 during the six transects. The interannual variation in $CH_4$ concentration during April in 2001, 2017 and 2018 at this salinity is a factor of 12. Secondly, the horizontal gradient in $CH_4$ concentration near the mouth of the Tamar estuary is very steep. Observations from April and July 2017 show a slope of between -20 and -50 nM per salinity unit. Salinity within the open water flux footprint varied between 32.2 and 35.2 between

September 2015 and August 2016, while salinity within the Plymouth Sound flux footprint varied between 32.0 and 35.1. The large range in $CH_4$ concentration and the strong and variable $CH_4$ vs. salinity relationship make any salinity-based prediction of

dissolved $CH_4$ concentration within the flux footprints highly uncertain. Thus we focus on estimating the transfer velocity of $CO_2$, but not $CH_4$ in the next section.

**3.4 $CO_2$ gas transfer velocity**

The implied $pCO_2$ from EC fluxes and in situ measured $pCO_2$ agree quite well over the annual cycle for the open water sector (Figure 7), suggesting that the use of the wind speed dependent transfer velocity parameterization of Nightingale et al. (2000) is largely reasonable in the mean. The variability in the implied $pCO_2$ (as indicated by the 25/75 percentiles), however, is sometimes greater than the variability in the in situ $pCO_2$. In this section, we estimate the time-varying $CO_2$ gas transfer velocity ($K_{CO2}$) and examine its variability and possible controls.

$K_{CO2}$ is computed as Flux $/\Delta pCO_2/sol_{CO2}$, where $sol_{CO2}$ is the solubility in $CO_2$. As shown in the previous section, $pCO_2$ measured from the open water flux footprint of PPAO is comparable to near-coincidental measurements at L4. Thus to estimate $K_{CO2}$ for the open water sector, we combine $pCO_2$ measurements from the open water footprint with the more numerous measurements at L4. To estimate $K_{CO2}$ for the Plymouth Sound sector, only $pCO_2$ measurements from that footprint are used. We linearly interpolate these seawater $pCO_2$ measurements to the times of the hourly $CO_2$ flux measurements. Interpolation more than four days away from the nearest $pCO_2$ observations is discarded. We chose four days (ca. half a week) here such that the computed $K_{CO2}$ values are retained if made between weekly $pCO_2$ measurements. The interpolated $pCO_2$ is then combined with the measured atmospheric $CO_2$ mixing ratio at PPAO to yield the air-sea $pCO_2$ difference ($\Delta pCO_2$). To normalize for the effect of temperature, $K_{CO2}$ is further adjusted to the Schmidt number of 660 ($K_{CO2,660} = K_{CO2} \bullet (660/Sc_{CO2})^{-0.5}$). The $CO_2$ solubility and Schmidt number as a function of temperature and salinity are taken from Wanninkhof et al. (2014). In order to minimize any bias in the computed $K_{CO2,660}$ due to the interpolation of daytime only $pCO_2$ measurements (see Section 3.2), we discard the nighttime (20:00 to 08:00 UTC) $K_{CO2,660}$ data during times of expected invasion (i.e. air-to-sea flux). The filtered hourly $K_{CO2,660}$ data are then averaged into 6-hour bins to reduce random noise.

**3.4.1 Dependence of $K_{CO2,660}$ on wind speed and friction velocity**

$K_{CO2,660}$ is plotted against the 10-m neutral wind speed ($U_{10n}$) in Figure 12, along with a $2^{rd}$ order polynomial fit. We have retained $K_{CO2,660}$ data here only when the absolute value of $\Delta pCO_2$ exceeded 20 µatm. This threshold is chosen as a balance between minimizing errors and maximizing data retention. A higher $|\Delta pCO_2|$ threshold (e.g. 40 µatm) does not obviously reduce the scatter in the $K_{CO2,660}$ vs. wind speed relationship. Error bars in $K_{CO2,660}$ are propagated from the standard errors in the fluxes. For the open water sector, $K_{CO2,660}$ shows a significant non-linear increase with wind speed ($R^2 = 0.35$, p<0.0001). The scatter in the $K_{CO2,660}$ and wind speed relationship is likely due to a combination of random uncertainties in the flux measurement (Yang et

al. 2016b) and variability in seawater $pCO_2$ not captured by the weekly measurements, as well as processes other than wind speed that affect gas exchange (see below). $K_{CO2,660}$ for the Plymouth Sound sector will be discussed in Section 4.3 within the context of bottom-driven turbulence.

The mean of the $K_{CO2,660}$ vs. wind speed relationship, as represented by the 2[rd] order polynomial fit, agrees (within a 95% confidence interval) with the widely-used relationship derived by Nightingale et al. (2000) using the dual tracer ($^3$He/SF$_6$) technique. We note that more recent parameterizations of the gas transfer velocity based on $^3$He/SF$_6$ and radiocarbon budgets (Ho et al. 2006; Sweeney et al. 2007; Wanninkhof 2014) are largely similar to Nightingale et al. (2000). In moderate-to-high winds, measured $K_{CO2,660}$ increases with wind speed at a rate that is less than cubic – a power fit yields an exponent of 1.3. This

is generally consistent with other recent closed-path EC $CO_2$ transfer velocity measurements (Butterworth and Miller 2016, Bell et al. 2017, Blomquist et al. 2017, and Landwehr et al. 2018).

At wind speeds less than ~5 m s$^{-1}$, measured $K_{CO2,660}$ at the PPAO are sometimes elevated and might not be entirely representative of air-sea transfer (Yang et al. 2016a). The EC friction velocity in the open water sector (see below and in Figure S2) is also at times higher than expected at these low wind speeds. The atmosphere was often more unstable at low wind speeds

($z/L$ of ~ -1), in part because low winds occurred more frequently during the warmer months. The Kljun et al. (2004) model predicts a flux footprint that is closer to the PPAO site during these conditions, such that the near shore environment (i.e. from the mast to the water's edge) might have some influence on the fluxes. Furthermore, the double rotation method used for the streamline correction of wind may be more uncertain at lower wind speeds. The planar fit method (Wilczak et al. 2001) could be superior under these conditions and will be an area of investigation during future analyses of PPAO flux data.

The friction velocity ($u_*$), a measure of air-sea total momentum transfer, is long thought to be a more direct representation of the drivers of turbulence and gas exchange than wind speed (e.g. Csanady et al. 1990). This appears to be the case especially for moderately soluble gases that are not significantly affected by bubble-mediated gas transfer, such as dimethyl sulfide (Huebert et al. 2010; Yang et al. 2011). The relationship between $K_{CO2,660}$ and the EC-derived $u_*$ shows a slightly better fit ($R^2 = 0.38$; Figure 13) than between $K_{CO2,660}$ and U$_{10n}$ ($R^2 = 0.35$). This is consistent with the idea that $u_*$ is a more suitable

predictor of $K$ than wind speed. The other benefit of relating $K_{CO2,660}$ with $u_*$ instead of U$_{10n}$ is that the $u_*$ measurement may be less affected by flow distortion than the U$_{10n}$ measurement (Landwehr et al. 2018).

The linear fit from Landwehr et al. (2018), derived from EC measurements of $CO_2$ flux in the Southern Ocean, is also shown in Figure 13. Compared to Landwehr et al. (2018), measurements at PPAO are similar at moderate $u_*$ values. At high $u_*$ values (strong winds), our estimates of $K_{CO2,660}$ increase with a greater power. Blomquist et al. (2017) demonstrated that waves

play a role in the open ocean air-sea exchange of $CO_2$ at high wind speeds and we expect waves to also influence $u_*$ (e.g. Edson et al. 2013). Waves shoal and steepen when they approach shallow water at the coast and generally break more frequently than

in the open ocean. $K_{CO2,660}$ measured at PPAO when waves are large might not be the same as $K_{CO2,660}$ over the open ocean. Unfortunately there were no wave measurements within the flux footprints to quantitatively investigate this effect .

**3.4.2 Seasonal variability in $K_{CO2,660}$**

We might expect the relationship between $K_{CO2,660}$ and wind speed to vary in different seasons due to the effects of bubbles and surfactants. Woolf et al. (1997) and Leighton et al. (2017) suggested an asymmetrical gas transfer rate that is faster for invasion than for evasion due to the hydrostatic pressure effect in bubble-mediated gas exchange, which is important for $CO_2$ (Bell et al., 2017; Blomquist et al. 2017). Figure 12 is color-coded by $\Delta pCO_2$ (positive when the ocean is supersaturated). We

see that invasion (i.e. air-to-sea) of $CO_2$, expected to occur in late spring and summer, was typically associated with low-to-moderate wind speeds. Evasion (i.e. sea-to-air) of $CO_2$, expected to occur in late autumn and winter, was typically associated with moderate-to-high wind speeds. There was limited overlap between invasion and evasion $K_{CO2,660}$ cases in the same wind speed range, partly due to gaps in the $pCO_2$ observations. Nevertheless, many of the $K_{CO2,660}$ data were well below the polynomial fit during periods of expected evasion and when $U_{10n}$ was between 6 and 10 m s$^{-1}$.

Recent measurements show large spatial and temporal differences in surfactant activity over the Atlantic Ocean (Sabbaghzadeh et al. 2017). A higher surfactant activity has been associated with suppression in the gas transfer velocity (e.g. Salter et al. 2011; Pereira et al. 2016, 2018). Figure 13 is color-coded by the near surface chlorophyll a concentration (Chl $a$), an indicator of phytoplankton biomass and biological activity. Chl $a$ was as low as 0.2 mg m$^{-3}$ in the winter and early spring, and as high as 5 mg m$^{-3}$ during late spring and summer. Many of the $K_{CO2,660}$ data for the open water sector were below the polynomial

fit at times of high Chl $a$ concentration. A seasonal variability in biologically-influenced surfactant activity seems likely and could alter the $K_{CO2,660}$ vs. wind speed relationship. Higher frequency observations of $pCO_2$ within the flux footprint (e.g. from a buoy) would greatly increase the number of transfer velocity estimates and enable a more robust comparison between invasion and evasion. Approaches similar to Sabbaghzadeh et al. (2017) and Pereira et al. (2016) on a seasonal scale, coupled with EC gas flux measurements, would help to address the importance of naturally-produced surfactants on gas exchange.


**3.4.3 Dependence of $K_{CO2,660}$ on bottom-driven turbulence**

Gas transfer driven by bottom-driven turbulence is parameterized as by Borges et al. (2004) as: $1.719\, v^{0.5}h^{-0.5}$ (cm hr$^{-1}$), where v is the current velocity (in cm s$^{-1}$), and h is water depth (in m). The authors treat this as a linearly additive term to wind driven gas exchange. For a depth of 10 m for the Plymouth Sound and a current velocity on the order of 1 m s$^{-1}$ during ebbing

and flooding tides (Siddorn et al. 2003), this leads to a transfer velocity as a result of bottom-driven turbulence of ~5 cm hr$^{-1}$ at a Schmidt number of 660. For the open water sector, gas transfer driven by bottom-driven turbulence is calculated to be less than

4 cm hr$^{-1}$ due to the deeper water. For reference, the Nightingale et al. (2000) parameterization at a wind speed of 6–9 m s$^{-1}$ is about 10–20 cm hr$^{-1}$. Thus bottom-driven turbulence may have a relatively large (~25%) influence on our observations of $K_{CO2,660}$ at low-to-moderate wind speeds. Neglecting bottom-driven turbulence could have resulted in overestimates when calculating implied GHG concentrations (Sections 3.1, 3.2), particularly at low wind speeds. Note though that our calculations of implied GHG concentrations were limited to wind speeds >5 m s$^{-1}$.

$K_{CO2,660}$ derived for the Plymouth Sound sector is also shown in Figures 12 and 13. Given the strong diurnal variability in the implied pCO$_2$ for this wind sector (see Figure 8), we have further limited $K_{CO2,660}$ to the time of day of 10:00 to 16:00 UTC. This strict filtering as well as the small number of coincidental flux and pCO$_2$ measurements result in only five 6-hour $K_{CO2,660}$ estimates for the Plymouth Sound sector. Plymouth Sound $K_{CO2,660}$ values roughly increase with wind speed and friction velocity, and are within the range of variability of the open water $K_{CO2,660}$ values. Note that four out of five of these $K_{CO2,660}$ estimates were associated with wind speeds over 9 m s$^{-1}$, for which bottom-driven turbulence is expected to have less influence (≤25% of the wind driven $K_{CO2,660}$). Future studies that combine EC flux measurements, frequent observations of seawater CO$_2$ and CH$_4$ concentrations within the footprint, and in situ measurements of current velocity would allow us to better test and improve $K$ parameterizations in shallow water.

### 3.5 Effects of rain on air-sea CO$_2$ exchange

Our year-long EC flux observations provide a valuable opportunity to directly assess the importance of rain on gas exchange. Mechanistically, rain could affect air-sea CO$_2$ flux through at least three mechanisms. First, lab studies show that the falling raindrops increase the near-surface turbulence, increasing total $K$ (e.g. Ho et al. 1997; Zappa et al. 2009). This effect is relatively more important at low wind speeds (e.g. Harrison et al. 2012). Secondly, rainwater could reduce near-surface pCO$_2$ via changes in the carbonate chemistry and gas solubility (e.g. dilution effect, Turk et al. 2010), and so result in more negative (or less positive) CO$_2$ fluxes. Lastly, dissolved CO$_2$ in rain droplets is taken up by the sea, which is often termed the wet deposition flux (e.g. Ashton et al. 2016). We examine each of these three mechanisms below.

*Effect on K:*

Figure S12 shows the hourly $K_{CO2,660}$ vs wind speed for the open water sector, color-coded by the measured precipitation rate at the surface ($P_s$). We use the hourly $K_{CO2,660}$ data here (filtered by a $|\Delta pCO_2| \geq 20$ µatm threshold) because rainfall is highly episodic. It is not obvious from our data that rain enhances $K$ at a given wind speed, which could be in part because typical rain rates at PPAO are roughly an order of magnitude lower than lab studies or parts of the tropics where rain rates are often tens of mm hr$^{-1}$. A caveat here is that the pCO$_2$ measurements were made approximately once a week from ~3 m depth. Thus they do

not fully describe short-term changes in pCO$_2$ at the air-sea interface as a result of rain. This could in turn influence the $K$ estimate.

*Dilution effect on near-surface pCO$_2$:*

To tease out the effect of rain on CO$_2$ flux via the dilution effect (and not on $K$), we focus on periods where we do not ordinarily expect to see much flux (i.e. when the expected $|\Delta pCO_2|$ is approximately zero). Figure S13 shows hourly CO$_2$ flux vs. rain rate for the open water wind sector. Here we have only retained data where the expected $|\Delta pCO_2|$ is within 10 µatm. Within our limited dataset and given the measurement uncertainties, it is not obvious that rain makes the CO$_2$ flux more negative (or less

positive) via the dilution effect. For the open water sector with $|\Delta pCO_2| \leq 10$ µatm, the mean CO$_2$ flux during rainy periods was - 5.3 (SE of 5.1) mmol m$^{-2}$ d$^{-1}$. During non-rainy periods, the mean CO$_2$ flux was -2.1 (SE of 2.1) mmol m$^{-2}$ d$^{-1}$. The two estimates are not statistically different from each other or from zero.

*Wet deposition flux:*

The wet deposition flux of CO$_2$ is estimated on an hourly basis as - $sol_{CO2} \cdot CO_{2,a} \cdot P_s$. Here it is assumed that the falling rain droplets are in equilibrium with atmospheric CO$_2$ (CO$_{2,a}$). The mean wet deposition flux over the entire year (including rainy and non-rainy periods) was computed to be about -0.1 mmol m$^{-2}$ d$^{-1}$, which is orders of magnitude smaller than the air-sea gas flux (e.g. Figure 5). During rainy periods, the mean wet deposition flux was -0.4 mmol m$^{-2}$ d$^{-1}$. Overall, the impact of rain on air-sea CO$_2$ exchange is fairly limited at PPAO, largely as a result of the modest rain rate.


**4 Conclusions**

Air-sea CH$_4$ and CO$_2$ fluxes measured by eddy covariance from a coastal location in the southwest UK over one year demonstrate significant variability on seasonal timescales. CH$_4$ flux in the coastal seas varied on a semi-diurnal (i.e. tidal) scale, while CO$_2$ flux at times varied diurnally. These observations suggest that sporadic samplings of seawater concentrations that are

limited to certain seasons, times of the day, or tidal cycle could result in biased annual mean flux estimates (see Sections 3.1 and 3.2). Surface ocean CH$_4$ saturations implied from the measured fluxes exceed a few hundred percent, and were higher over the semi-enclosed Plymouth Sound than over open water. These results are consistent with the trend in dissolved CH$_4$ concentration observed from the upper part of the river Tamar to the mouth of the Plymouth Sound. The coastal sea was a net sink of CO$_2$ in late spring and summer, and a net source of CO$_2$ in autumn and winter. CO$_2$ flux from the Plymouth Sound demonstrated greater

diurnal variability than the CO$_2$ flux from the open water sector. We estimate the CO$_2$ transfer velocity from our measurements of fluxes and in situ seawater concentrations. The mean derived CO$_2$ transfer velocity at this coastal location agrees reasonably

well with previous tracer-based and closed-path $CO_2$ eddy covariance estimates from the open ocean. Rainfall does not appear to have a large direct effect on air-sea $CO_2$ exchange at our temperate coastal site. There are hints of seasonality in the transfer velocity vs. wind speed relationship that may be related to asymmetric bubble-mediated gas exchange or biologically derived

surfactants. The effect of bubbles, surfactants and bottom-driven turbulence warrant further investigation in order to improve understanding of air-sea gas exchange and estimates of coastal Greenhouse Gas budgets.

**Author Contribution**

MY and TB designed and performed the flux measurements and data analysis/interpretation. IB and AR measured dissolved

$CH_4$ concentration, while VK measured seawater $pCO_2$. JF and TS supplied the underway and buoy data from the Western Channel Observatory. TS and PN provided helpful comments on the focus and context of the paper.

**Acknowledgment**

This work contributes to the ACSIS (The North Atlantic Climate System Integrated Study; NE/N018044/1), MOYA (Methane

Observations and Yearly Assessments; NE/N015932/1), LOCATE (Land Ocean Carbon Transfer; NE/N018087/1), and CLASS (Climate Linked Atlantic Sector Science) projects funded by the Natural Environment Research Council (NERC), UK. The Western Channel Observatory is funded by NERC's National Capability programme. Trinity House (http://www.trinityhouse.co.uk/) owns the Penlee site and has kindly agreed to rent the building to PML so that instrumentation can be protected from the elements. We are able to access the site thanks to the cooperation of Mount Edgcumbe Estate

(http://www.mountedgcumbe.gov.uk/). We thank the Environmental Agency for the Tamar flow data. We also thank F. E. Hopkins (PML), M. J. Yelland (National Oceanography Centre), I. M. Brooks (University of Leeds), and J. Prytherch (Stockholm University) for continued measurement support. This is contribution number 5 from the Penlee Point Atmospheric Observatory.

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

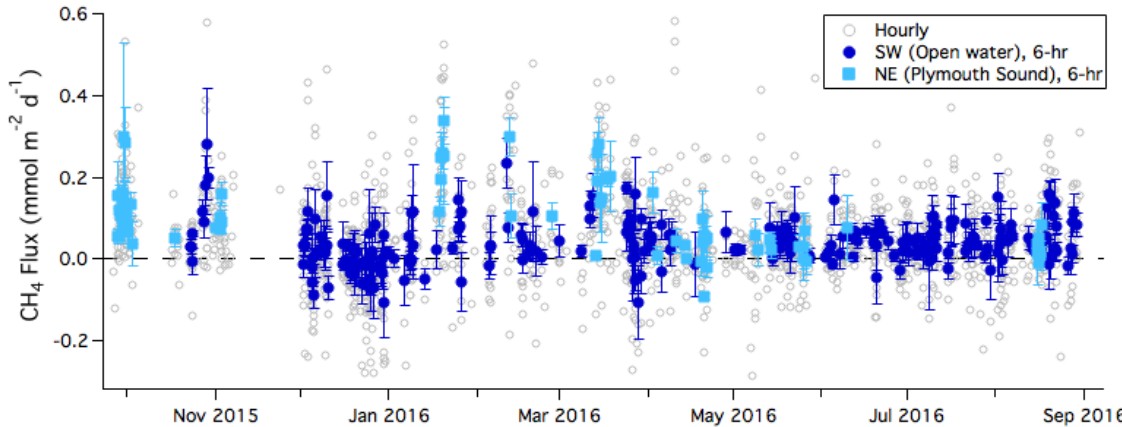

Figure 1. One-year time series of CH$_4$ flux (hourly average) during times when winds were from the sea. Six-hour averages of fluxes are further separated into the southwest (open water) and northeast (Plymouth Sound) wind sectors. Error bars indicate standard error.

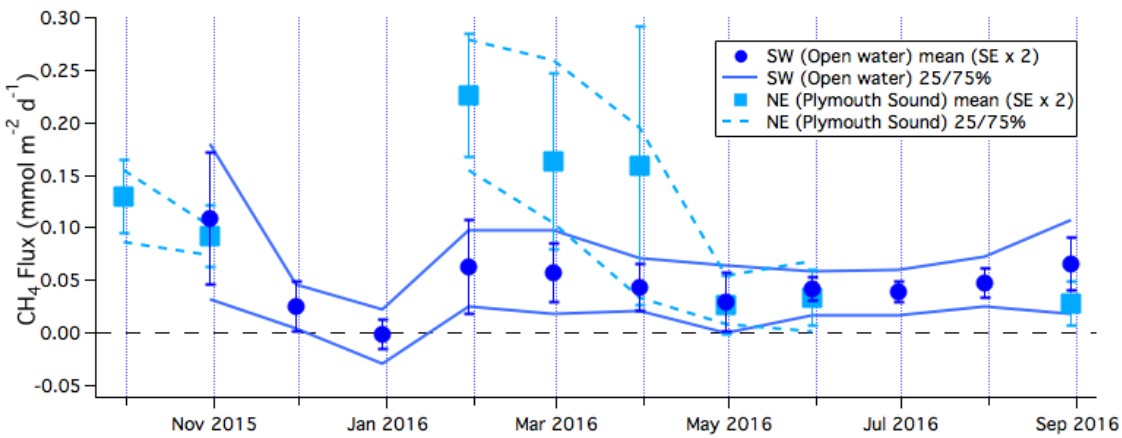

Figure 2. Monthly averages and 25/75 percentiles of CH$_4$ flux from the southwest (open water) and northeast (Plymouth Sound) wind sectors. Error bars indicate 2 times standard error.

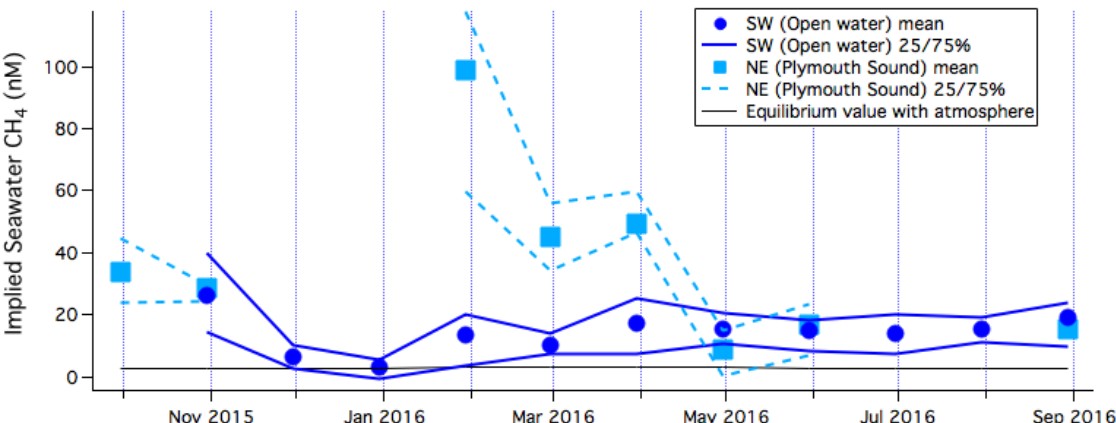

Figure 3. Monthly averages and 25/75 percentiles of implied CH$_4$ concentration for the southwest (open water) and northeast (Plymouth Sound) wind sectors, along with the equilibrium value with respect to the atmosphere.

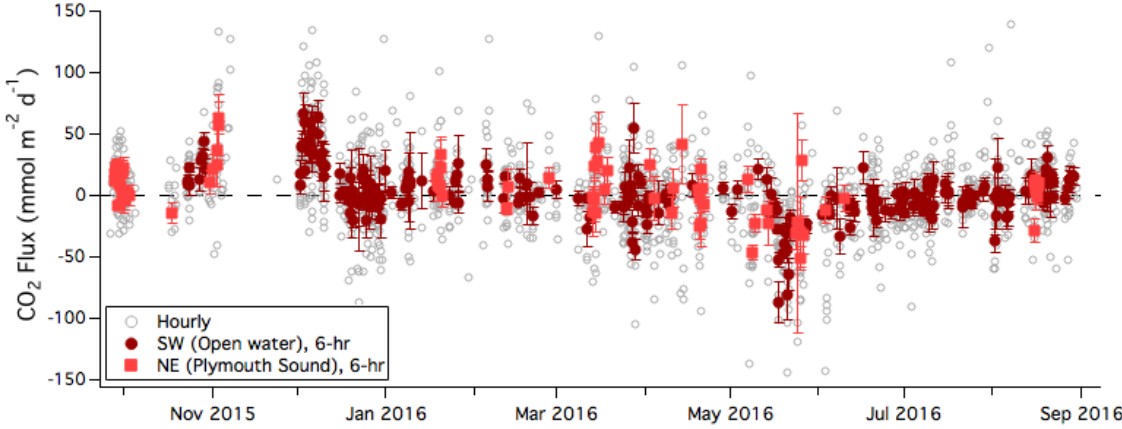

Figure 4. One-year time series of $CO_2$ flux (hourly average) during times when winds were from the sea. Six-hour averages are further separated into the southwest (open water) and northeast (Plymouth Sound) wind sectors. Error bars indicate standard error.


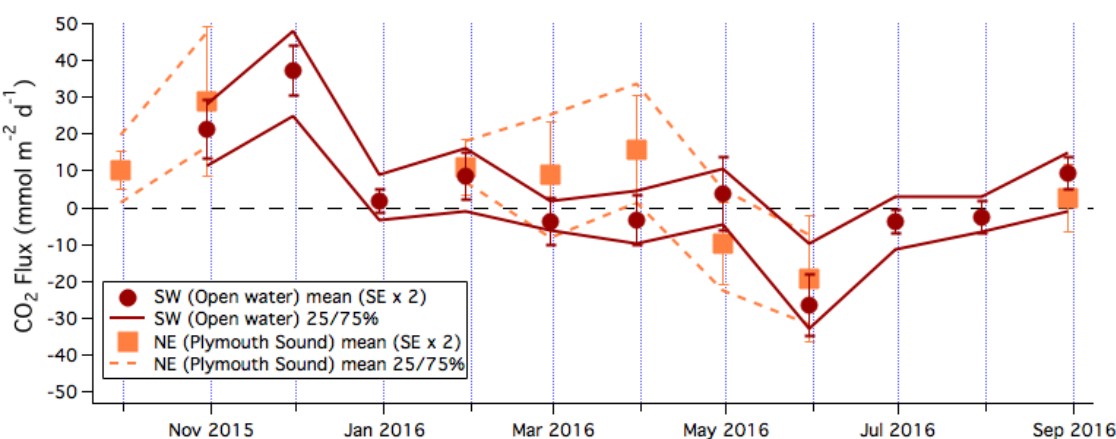


Figure 5. Monthly averages and 25/75 percentiles of $CO_2$ flux from the southwest (open water) and northeast (Plymouth Sound) wind sectors. Error bars indicate 2 times standard error.

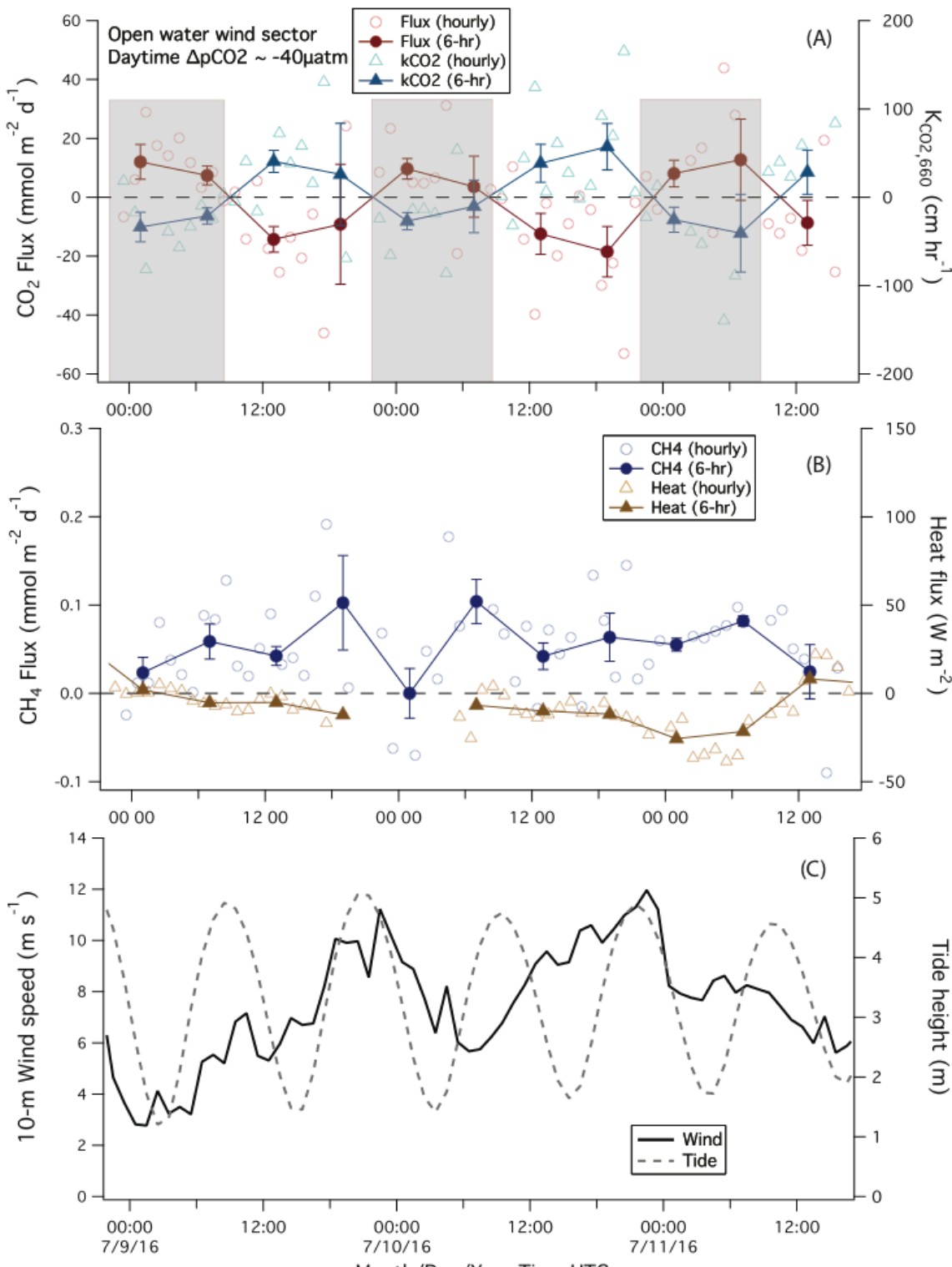

Figure 6. Example of variability in: A) $CO_2$ flux and computed transfer velocity; B) $CH_4$ flux and sensible heat flux; and C) wind speed and tidal height during a period of southwesterly winds. Fluxes are shown in both hourly and 6-hr averages. Note that the negative transfer velocity ($K$) values at night (shaded) computed from the measured $CO_2$ flux and interpolated daytime $pCO_2$ are non-physical, and likely due to unaccounted for diurnal variability in seawater $pCO_2$.



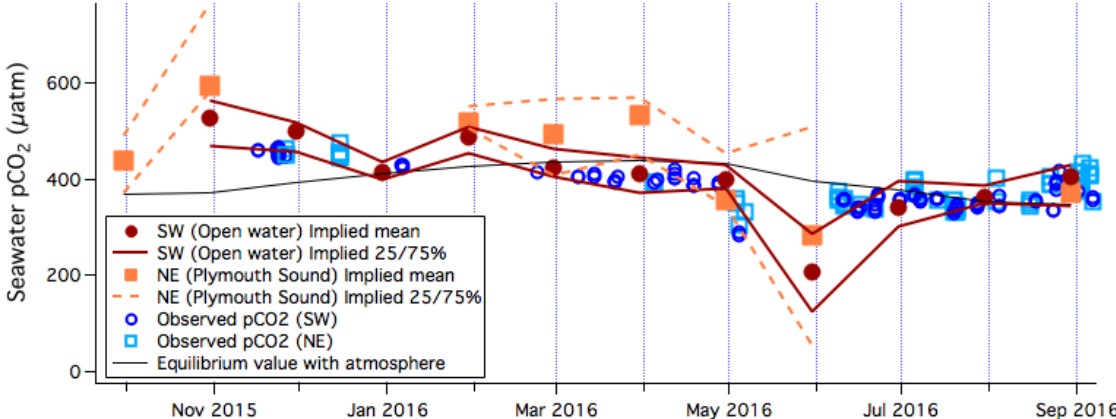

Figure 7. Monthly averages and 25/75 percentiles of implied seawater $pCO_2$ for the southwest (open water) and northeast (Plymouth Sound) wind sectors. Observed $pCO_2$ from the Plymouth Quest within the southwest sector (plus at L4) and within the northeast sector are also shown, along with the equilibrium value with respect to the atmosphere.


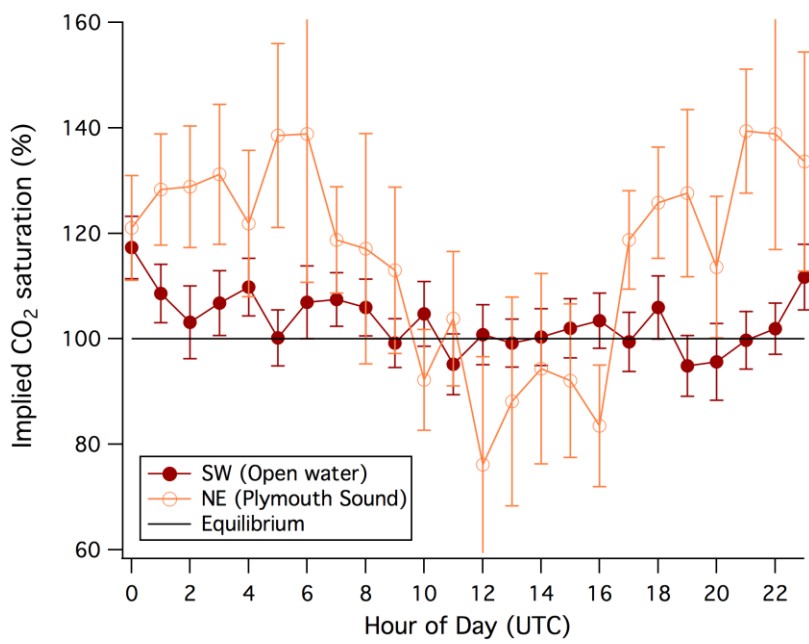

Figure 8. Mean diurnal variability in the implied seawater saturation of $CO_2$ for the for the southwest (open water) and northeast (Plymouth Sound) wind sectors. Error bars indicate standard errors.


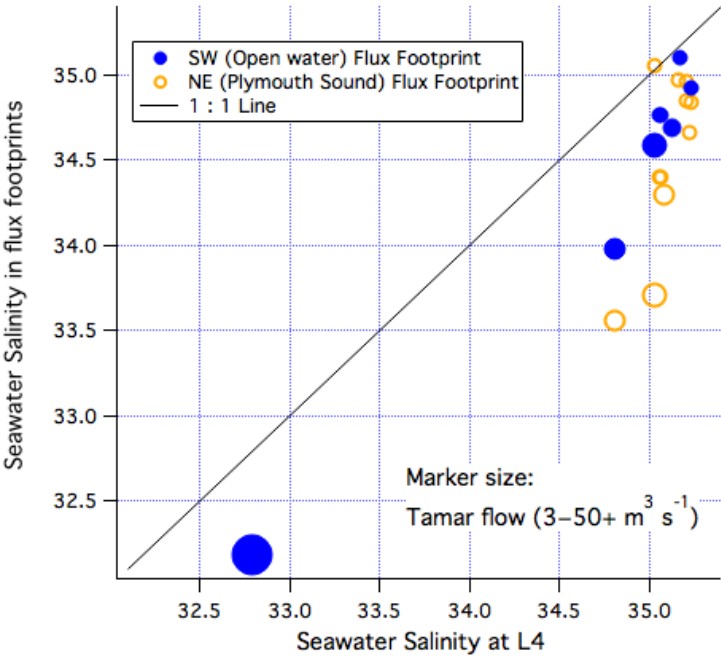

Figure 9. Salinity measured within the two air-water flux footprints of Penlee vs. near-coincidental measurements from the *Quest* at the L4 station. The size of the markers corresponds to the flow rate in the Tamar river, as measured at Gunnislake.


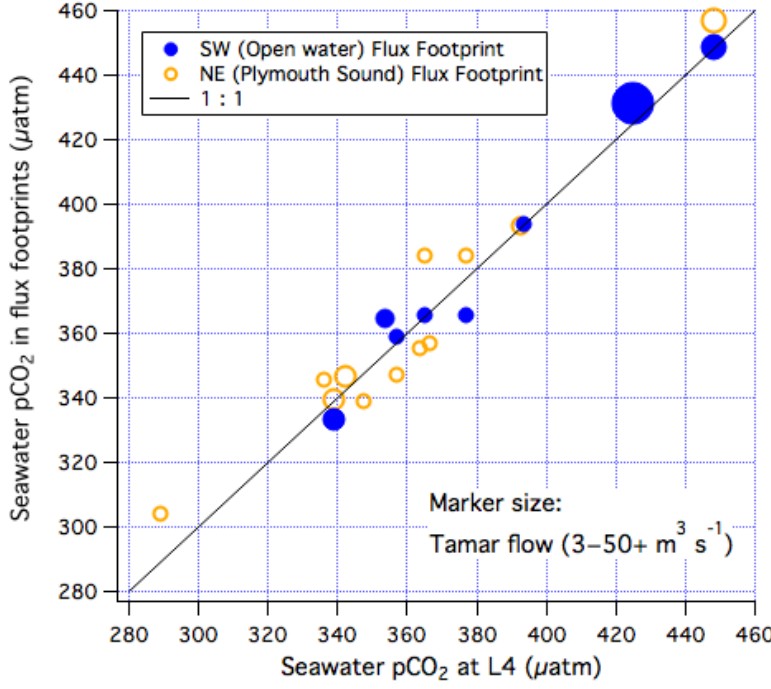

Figure 10. pCO$_2$ measured within the two air-water flux footprints of Penlee vs. near-coincidental measurements from the *Quest* at the L4 station. The size of the markers corresponds to the flow rate in the Tamar river, as measured at Gunnislake.


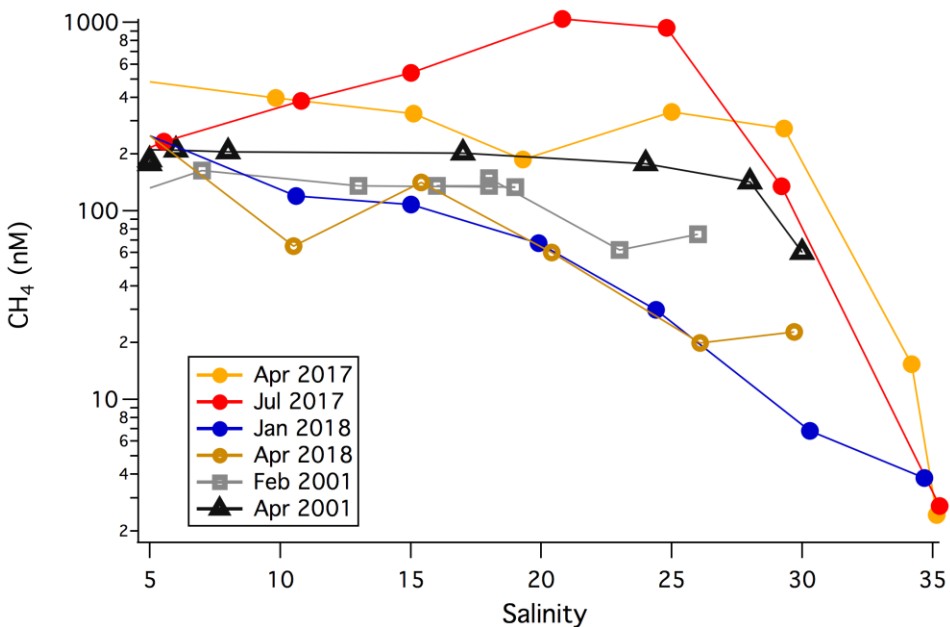

Figure 11. Dissolved $CH_4$ concentration from the Tamar river to the L4 station varies with salinity. Data from 2017 and 2018 were made during LOCATE sampling. The 2001 data are taken from Upstill-Goddard et al. (2016).

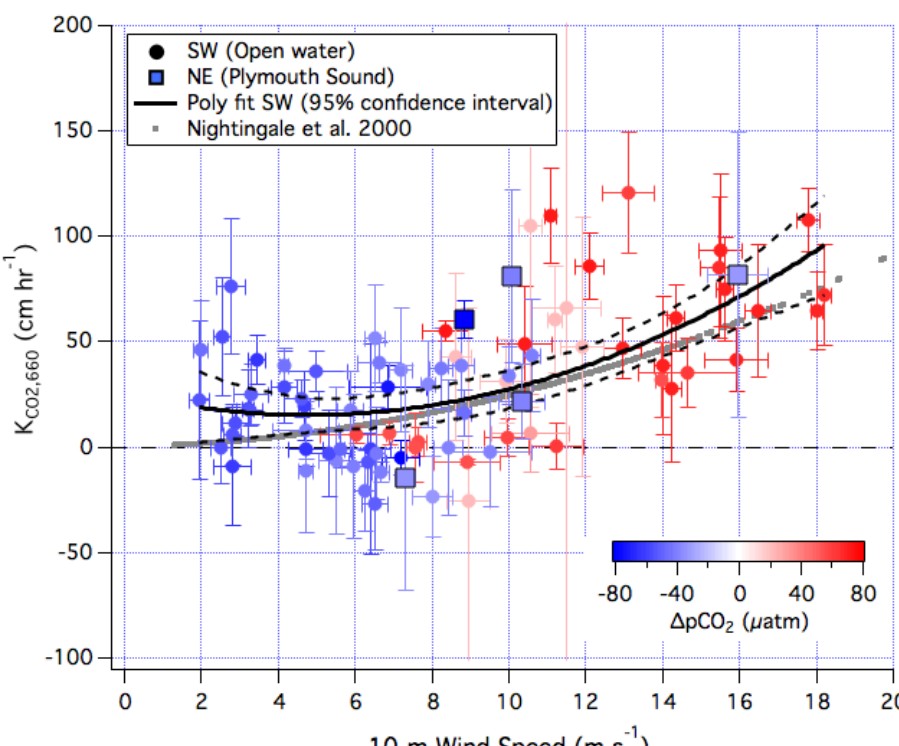

Figure 12. $CO_2$ transfer velocity (normalized to a Schmidt number of 660) vs. 10-m neutral wind speed for both the southwest (open water) and northeast (Plymouth Sound) wind sectors. Note that color-coding is capped at $|\Delta pCO_2|$ values of 80 µatm for clarity.

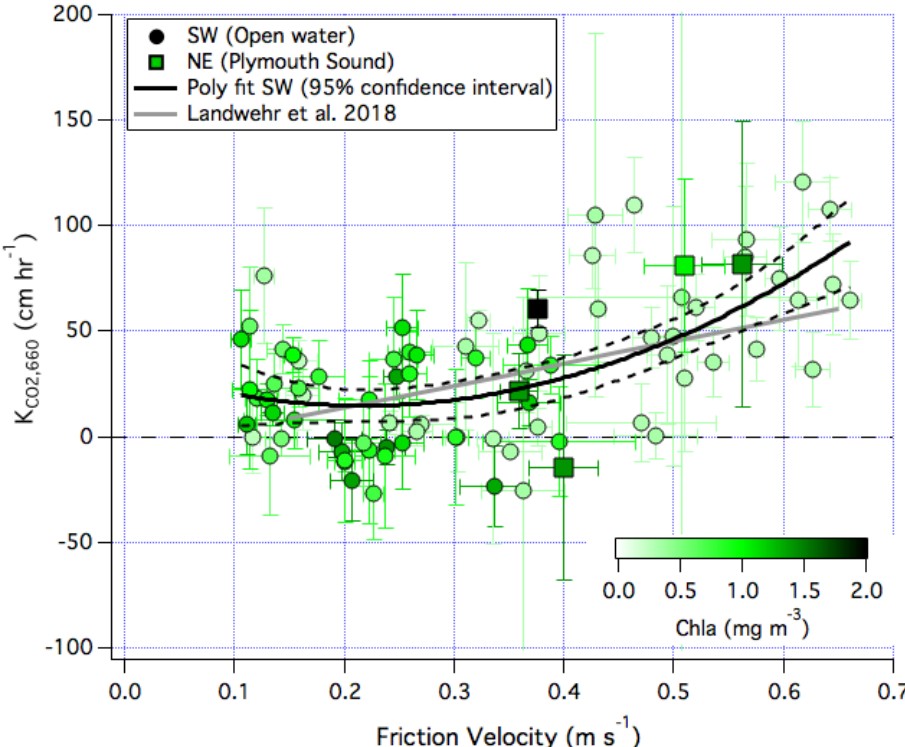

Figure 13. $CO_2$ transfer velocity (normalized to a Schmidt number of 660) vs. the friction velocity for both the southwest (open water) and northeast (Plymouth Sound) wind sectors. Note that color-coding is capped at a chlorophyll a concentration of 2 mg $m^{-3}$ for clarity.
