# Peer review of "Insights from year-long measurements of air-water CH4 and CO2 exchange in a coastal environment"

_Biogeosciences, 2018_

## Short Comment (SC1) · 18 Dec 2018

Please see the Supplement pdf document.

Please also note the supplement to this comment:
https://www.biogeosciences-discuss.net/bg-2018-503/bg-2018-503-SC1-supplement.pdf

---

## Referee Comment (RC1) · Anonymous Referee #1 · 21 Jan 2019

- General comments: this manuscript by Yang et al deals with CH4 and CO2 fluxes in a coastal environment. Assessing CO2 and CH4 air-water exchanges is an important exercise to determine the impact of given ecosystems on the atmospheric CO2 and CH4 burden. It is particularly the case for aquatic ecosystems such as estuarine and coastal ones which are of relative influence compared to the area they are covering at the global scale. Most of the previous studies dealing with the subject have been based on indirect estimate through air-sea concentration difference and gas transfer velocity, the so-called Boundary-Layer method. The work by Yang et al presents an interesting and rather rare time series of EC measurement performed over one year. The authors have done a good job in data collecting and study design at the Penlee Point Atmospheric Observatory (PPAO), on a nearby buoy (L4), and from different research

Vessels. Data base includes $CO_2$ and $CH_4$ exchange fluxes as well as a description of meteorological data and some of the water quality parameters (Chla for example). This MS is generally well written, is timely and interesting to understand the parameters of influence on $CO_2$ and $CH_4$ exchanges in coastal environments. Several parameters of influence on transfer velocities have been checked, all of them are relevant. Though, curiously, the effect of precipitation rate on fluxes have not been investigated. Impact of drops on the water surface can enhance significantly (several tens of percent) the gas transfer velocity. Were the precipitation periods withdrawn from the date as part of the EC quality control process? In all cases, the influence of precipitation of the data set (whether on the EC data quality or on the transfer velocity) should be discussed.

As pointed out by Nilsson and colleagues, statement by Yang and colleagues on the performance of open-path sensor should be revised. Sentences should be reworded to include a more tempered statement on potential interferences of open-path analyzer over water bodies. Effect of salinity on these spectral interferences should be discussed as suggested by Nilsson.

- Specific comments: here are some specific comments that should strengthen the MS.

P5, l 133: Can you quantify more precisely the effect of stability on the Xmax and X90 distances? This would help for the discussion on $CO_2$ fluxes on p8

P7, l 185-195: Not clear, mean flux should be the same whatever the way it is calculated.

P7, l 192: not clear, but 6h fluxes should be the reference fluxes when compared to annual fluxes, how could they be skewed?

P8, l 212: give details on how the total $CH_4$ flux was calculated

P8, l 216: give details to the reader on how the random instrument noise is calculated. Is the instrumental noise mentioned on line 219 the same noise?

P8, l 229: only daytime measurements of $pCO_2$ are mentioned, no night time measurements performed, right?

P8, l 231: Not clear which data were interpolated, and how they were interpolated

P8, l 235: see comment on page 5. How far further upwind?

P9, l 248-29: again, not clear why mean calculated from monthly mean and from 6h mean are (so) different

P9, l 259: there are many speed-dependant transfer velocity relationships. Choice of the only one from Nightingale et al 2000 paper should be justified. On which basis this choice was made.

P9, l 261: wind speed threshold above 5 ms-1 seems quite high. Any justification of that wind speed value?

P9 l 264-268: saturation level relative to atmospheric saturation are defined but not used on figure 6. This could be done for the reader to better follow the discussion

P9, l 268: Is the effect of salinity and temperature accounted for in the 14% variation of CH4 solubility? Not evident on figure 6.

P10, l 277: which time series is commented here, 6h or 1h mean data?

P10, l278: same pattern that what? Semi-diurnal variability? That is not possible, this must be something else. . .

P10, l 285-287: comparison is made on two set of data without the same number of monthly data. Not sure it is meaningful.

P11, l 299-300: seems that the sentence should be reworded

P11, 309: syntax? Missing word?

P12, l 350: not the highest saturation, but highest absolute concentration.

P12, l 353-354: data at L4 not on figure, could be added for comparison.

P12, l 357: four "times" higher. Times is missing

P13, l 359: mentioned that it is the flux footprints for the open water sector

P13, l 383: how did you choose to discard interpolation more than four days away?

P14 l 393: how this threshold of 20 $\mu$atm was chosen/determined? Any justification?

P14, l 397: be more explicit on what you call "measurements uncertainties in flux, variability in pCO2, as well as processes other than wind speed . . ."

P14, l 413-414: why did not you check the effect of planar fit vs double rotation to confirm your assumption?

P15, l 441: what is the effect of the 40$\mu$atm threshold on that gap?

P15 : the highest Chla measurements in Plymouth Sound is well above the regression line. Any clue for that?

P14, l 467: Be more clear about "to reduce the temporal mismatch between the flux and pCO2 measurements".

P15, l 479: ". . .could result in biased annual mean flux estimates". Could this be more precisely quantified for example in the case of only daytime measurements?

Figures P23: use consistent legend throughout the paper with: SW (open water) and NE (Plymouth Sound) all along.

P25: figure 5A: you should not display negative value for K which have no physical meaning. You could shade the night time period on the figure.

P27: figure 9: add units to the X and Y axis

---

## Short Comment (SC2) · Dear Nilsson et al. · 21 Jan 2019

Thanks a lot for your comment. We were not aware of the Nilsson et al. 2018 paper, and we are very glad that you undertook a close examination of the potential effect of water vapor on the Licor CO2 fluxes. It is indeed interesting that you did not observe a large difference in CO2 flux between the open path Li7500 and the dried closed path Li7200 on average at your land site and the Baltic Sea site, in contrast to previous oceanic studies. In light of this, we agree to remove the following sentence from the BGD paper.

"Unfortunately the open-path sensors utilized at OÌLstergarnsholm and Punta Morro

(LI-7500, LI-COR Biosciences) could well be affected by $CO_2$-$H_2O$ spectral interference (Blomquist et al. 2014; Landwehr et al. 2014; Butterworth and Else, 2018), likely resulting in biased fluxes under conditions of significant latent heat flux."

Out of scientific interest, your paper suggests that the insensitivity in $CO_2$ fluxes at these sites towards $H_2O$ could be due to the low abundance of sea spray, and the use of the variance in relative signal strength index (RSSI) effectively removes any spurious fluxes. Figure 3 and Figure A1 in your paper clearly illustrate that the dried and undried $CO_2$ fluxes strongly diverge when variance in RSSI >1e-3. What do you think causes the variability in RSSI in the dried Licors (presumably not water droplets or sea spray)? Are previous gas exchange studies from Ostergarnsholm also filtered with this same RSSI threshold? We are happy to continue this discussion offline, as it is unrelated to the current BGD paper.

Sincerely,

Mingxi Yang

---

## Referee Comment (RC2) · Anonymous Referee #2 · 28 Jan 2019

General Comments

Yang et al. present an annual monitoring of CH4 and CO2 fluxes from a coastal location in the southwest UK, using the eddy covariance technique. The paper is generally well-written and easy to follow. The manuscript shows important findings related to the investigation of carbon dynamics in coastal zones, which remains largely unknown due to its intrinsic high spatio-temporal variability. The authors showed high differences between fluxes measured at daytime and nighttime, implying that results covering only daytime may be biased due to the influence of biological activities (called diurnal) and tidal processes (called semi-diurnal). The methodology of in situ measurements and data processing are consistent. However, the authors must to address the important appointments from Erik Nilsson and the other anonymous referee. The authors also

must improve the description of state of art of CO2 and CH4 dynamics in coastal zones and estuaries (introduction). This section is poorly described. The same appointment is also true for the discussion section related to the dynamics of these GHGs. This part of the manuscript is also unsatisfactorily constructed. The manuscript is well described/structured in terms of technical aspects, but the data interpretation is not sufficient discussed in terms of ecological/biogeochemical processes. A more detailed bibliographical survey is strongly recommended to support your findings and to better contextualize this study.

Specific Comments

Abstract What it means the semi-diurnal timescales (tidal processes)? Please, describe what is the pCO2 (partial pressure of CO2 in water/air).

Introduction

See the general comments.

Lines 31-32: "...have been increasing over the last few hundred years primarily due to human activities (Hartmann et al. 2013)." I agree. However, the fastest increase is related to the last 50 years... Lines 38-43: This paragraph is a simplistic exposition of the CH4 dynamics/cycling. You should go deep in this part, especially in the studies covering the coastal oceans/estuaries. I recommend a better literature research.

Lines 45-47: "Globally averaged, the open ocean is modelled to absorb about a quarter of the anthropogenic CO2 emission (Le Quéré et al. 2015)." You can give numbers, and update this reference (from the global carbon budget 2018; Le Quéré et al. 2018). https://www.earth-syst-sci-data.net/10/2141/2018/

Lines 50-56: "Estuaries, on the other hand, are generally net sources of CO2 for the atmosphere (e.g. Frankignoulle et al. 1998)..." How much? Frankinoulle et al. 1998 is a classical reference. However, there are more recent references that you should include when is describing the global emissions of CO2 by estuaries. In addition, you

should shortly describe processes that can affects the dynamics and fluxes at the air-water interface.

Lines 55-56: "The shallow seas are predicted to become a greater net sink of CO2 in the future due to rising atmospheric CO2 and increasing inorganic nutrients (e.g. Andersson and Mackenzie, 2004)." However, other studies showed that the estuaries could emit more CO2 due to the enhancing of organic matter respiration.

Lines 73-74: "Thus a wind speed-only dependent representation of K, incomplete for the open ocean (Wanninkhof et al. 2009)." Please rephrase.

Experimental Lines 105-107: "For this paper, wind data from the Windmaster Pro sonic anemometer were used between September 2015 and March 2016. Since March 2016, wind data from the R3 sonic anemometer (not operational for the first 6 months of this annual study) were preferred because of its higher precision and better performance during heavy rain events. Did you compare the wind velocity from the 2 different anemometers used in this study?

Results Fig. 1. Could you better explain the causes of the negative fluxes of CH4? This means that the water was sub-saturated with respect to atmospheric CH4 concentrations? What are the main causes of this under saturation?

Could you add a plot combining the results of chl a concentrations and the pCO2 fluxes/values (scatter plot)?

Lines 210-213: "If the PPAO open water footprint is representative of the nearest 1.4 km (i.e. X90 of our fluxes, see Section 2.2) of the UK coast, our measurements extrapolate to a total CH4 flux of 4.8 Gg yr-1 in UK coastal seas." The extrapolation of the results to other areas is a good exercise. However, CH4 is a gas that present very special conditions of production and consumption. I am not convinced about this calculation.

Lines 219-221: "Ambient variability (in e.g. dissolved concentrations) largely drives the rapid temporal fluctuations in CO2 flux, which is unlikely to be fully captured by weekly

or monthly seawater sampling." This passage is confuse. Please rewrite.

Fig. 4. CO2 flux from the Plymouth Sound sector appeared to be more positive than from the open water sector in some months. I would expect larger differences, but this is not the case. Could you explain this?

Lines 239-242: "Wind speed was generally higher at night during these few days and the measured fluxes imply that $\Delta$pCO2 (see next sections on this calculation) changed from about -40 $\mu$atm during the day to about 15 $\mu$atm at night." Then, during this period, the system was a net sink of CO2?

Lines 283-285: "The greatest undersaturation in CO2 is observed in late spring and early summer, coinciding with an increase in chlorophyll a concentration at the nearby L4 station (Figure S6)." As I said before, I would like to see graph showing the chl a and the CO2 concentrations.

Lines 295-298: "The diurnal variability we observed is important in the context of estuarine CO2 (and carbonate system) observations that are predominantly carried out during daytime when sampling and navigation are easier." Many studies were published recently covering the diurnal (biological effect) and the semi-diurnal (tidal effect) variability on pCO2 changes, which are poorly described and constrained in the present manuscript.

Line 304: Spatial variability in seawater concentrations. Please see the general comments. This could be because the influence of Tamar estuary on the PPAO flux footprint is less in terms of pCO2 (e.g. due to the already large burden of carbonate and bicarbonate in seawater), and more on physical and biogeochemical processes. I am not sure about this statement.

Line 306: "dissolved pCO2." This is unusual. You should refer "pCO2" or "dissolved CO2".

Lines 329-331: "It could be that the large burden of carbonate and bicarbonate in

seawater partially buffered the impact of the freshwater input on pCO2 within the flux footprints." This is poorly discussed.

Lines 373- 377 "The implied pCO2 from EC fluxes in monthly bins and in situ measured pCO2 agree quite well over the annual cycle for the open water sector (Figure 7), suggesting that the use of the wind speed dependent transfer velocity parameterization of Nightingale et al. (2000) is largely reasonable." I am not sure that this type o graph is the best to show a comparison between estimated and measured pCO2.

Figure 7. Y-axis : seawater pCO2 ?

[Figure]

---

## Author Comment (AC1) · 12 Feb 2019

Author Comment with regard to:

"Insights from year-long measurements of air-water CH4 and CO2 exchange in a coastal environment"

by Yang et al.

Many thanks for the detailed *comments and suggestions from Anonymous Referee #1*. We are very glad to hear that the referee found our contribution timely and interesting. Below are our replies to the referee's comments, which are in *italic*.

***Anonymous Referee #1***
*General comments: this manuscript by Yang et al deals with CH4 and CO2 fluxes in a coastal environment. Assessing CO2 and CH4 air-water exchanges is an important exercise to determine the impact of given ecosystems on the atmospheric CO2 and CH4 burden. It is particularly the case for aquatic ecosystems such as estuarine and coastal ones which are of relative influence compared to the area they are covering at the global scale. Most of the previous studies dealing with the subject have been based on indirect estimate through air-sea concentration difference and gas transfer velocity, the so-called Boundary-Layer method. The work by Yang et al presents an interesting and rather rare time series of EC measurement performed over one year. The authors have done a good job in data collecting and study design at the Penlee Point Atmospheric Observatory (PPAO), on a nearby buoy (L4), and from different research Vessels. Data base includes CO2 and CH4 exchange fluxes as well as a description of meteorological data and some of the water quality parameters (Chla for example). This MS is generally well written, is timely and interesting to understand the parameters of influence on CO2 and CH4 exchanges in coastal environments. Several parameters of influence on transfer velocities have been checked, all of them are relevant. Though, curiously, the effect of precipitation rate on fluxes have not been investigated. Impact of drops on the water surface can enhance significantly (several tens of percent) the gas transfer velocity. Were the precipitation periods withdrawn from the date as part of the EC quality control process? In all cases, the influence of precipitation of the data set (whether on the EC data quality or on the transfer velocity) should be discussed.*

This is an interesting question. Rain can add substantial noise to the measurement of wind velocities by sonic anemometers, especially at high frequencies, thereby increasing the uncertainty in the EC fluxes. In our case, the Gill R3 anemometer (used for the second half of the 1-yr campaign) is much less affected by rain than the Windmaster Pro anemometer (used for the first half of the 1-yr campaign) for CO2 flux. We have not filtered our CO2 flux by a threshold rain rate (e.g. a commonly used value would be 1 mm/hr), but our quality control filtering that includes the noise level of the vertical wind velocity has removed most of the fluxes during rainy periods for the first half of the 1-yr campaign. The second half of the 1-yr campaign (when the R3 anemometer was used) retains more rainy periods.

Would our annual mean CO2 flux be significantly biased by excluding most of the rainy periods? Over the entire year, hourly rain rate exceeded 1 mm/hr 7% of the time at PPAO when winds were from southwest. The average wind speed was about 50% higher during these rainy periods than during non-rainy periods. The Nightingale et al. (2000) wind speed relation predicts that the average K (and so flux magnitude) should be about 120% higher during rainy periods than during non-rainy periods. Thus excluding fluxes during these rainy periods could result in an underestimation of the mean annual flux by approximately 8.4% (=0.07*1.2).

Mechanistically, rain could affect direct air-sea CO2 flux through at least three mechanisms. First, lab studies show that the falling raindrops increase the near-surface turbulence, thereby increasing total K (e.g. Ho et al., 1997; Zappa et al., 2009). This effect is relatively more important at low wind speeds (e.g. Harrison et al., 2012). Secondly, rainwater could reduce the near-surface pCO2 via changes in the carbonate chemistry and gas solubility (e.g. dilution effect, Turk et al. 2010), and so result in more negative (or less positive) CO2 fluxes. Lastly, dissolved CO2 in rain droplets are taken up by the sea, which is often

termed the wet deposition flux (e.g. Ashton et al. 2016). We examine each of these three mechanisms below:

1. Effect of rain on K

The plot below shows the hourly kCO2,660 vs wind speed for the open water sector, color-coded by rain rate (data filtered by a |dpCO2|≥20 uatm threshold). We use the noisier hourly KCO2,660 data here because rainfall is highly episodic. It is not obvious that at a given wind speed, rain enhances K. This could be in part because compared to lab studies or parts of the tropics where rain rates are often on the order of tens of mm/hr, the typical rain rates at PPAO are roughly an order of magnitude lower. A caveat here is that the pCO2 measurements were made approximately once a week from ~3 m depth. Thus they do not fully describe short-term changes in pCO2 at the air-sea interface as a result of rain. This could in turn influence the K estimate.

[Figure]

2. Dilution effect (changes in near-surface pCO2)

To tease out the effect of rain on CO2 flux via the dilution effect (and not on K), we focus on periods where we do not ordinarily expect to see much flux (i.e. when the expected |dpCO2| is approximately zero). The plot below shows hourly CO2 flux vs. rain rate for the open water wind sector. Here we have only retained data where the expected |dpCO2| is ≤10 uatm. Within our limited dataset and given the measurement uncertainties, it is not obvious that rain makes CO2 flux more negative (or less positive) via the dilution effect at PPAO. For the open water sector with |dpCO2| ≤10 uatm, the mean CO2 flux during rainy periods was -5.3 (SE of 5.1) mmol/m2/d. During non-rainy periods, the mean CO2 flux was -2.1 (SE of 2.1) mmol/m2/d. The two estimates are not statistically different from each other as well from zero.

[Figure]

3. Wet deposition flux

The wet deposition flux of CO2 is estimated on an hourly basis as - $sol_{CO2}$ * $CO_{2,a}$ * rain_rate.  Here it is assumed that the falling rain droplets are in equilibrium with the atmosphere in terms of $CO_2$.  The mean wet deposition flux over the entire year (including rainy and non-rainy periods) was computed to be about -0.1 mmol/m2/d, which is orders of magnitude smaller than the air-sea gas flux (e.g. Figure 4).  During rainy periods only, the mean wet deposition flux was -0.4 mmol/m2/d.  Overall, we see that the impact of rain on air-sea $CO_2$ exchange is fairly limited at PPAO, largely as a result of the modest rain rate.

We will add the discussion above to a newly created section 6 and to the supplementary materials of the manuscript.

*As pointed out by Nilsson and colleagues, statement by Yang and colleagues on the performance of open-path sensor should be revised. Sentences should be reworded to include a more tempered statement on potential interferences of open-path analyzer over water bodies. Effect of salinity on these spectral interferences should be discussed as suggested by Nilsson.*

Per suggestion by Nilsson et al., we have agreed to remove the sentence about the performance of the open path sensors from our manuscript.

*- Specific comments: here are some specific comments that should strengthen the MS.*
*P5, l 133: Can you quantify more precisely the effect of stability on the Xmax and X90 distances? This would help for the discussion on CO2 fluxes on p8*

The effect of stability on the footprint for this site was discussed in Yang et al. 2016a.  In the Kljun et al. 2004 model, stability is simply accounted for by adjusting the ratio between the standard deviation in w ($\sigma$w) and the friction velocity u$_*$.  We represented the strongly unstable, neutral, and strongly stable cases with $\sigma$w: u$_*$ ratios of 1.75, 1.3, and 0.9, respectively.  Compared to a neutral atmosphere, Xmax (as well as X90) is about 20% closer to PPAO under the unstable case and 35% further away from PPAO under the stable case above.  We will refer to our 2016 paper in the BGD manuscript.  Note that the strongly unstable and stable conditions above correspond to air-sea sensible heat fluxes > ~200 W/m2 and <~ -100 W/m2, respectively.  Sensible heat flux measured at Penlee Point had typical magnitudes of tens of W/m2 (see line 173) and thus the atmosphere was generally quite close to neutral.

*P7, l 185-195: Not clear, mean flux should be the same whatever the way it is calcu- lated.*
*P7, l 192: not clear, but 6h fluxes should be the reference fluxes when compared to annual fluxes, how could they be skewed?*

The number of valid flux measurements, largely depending on the wind direction, varied from month to month. In some months there were valid flux measurements for the open water sector ~40% of the time (i.e. ~12 days), and in other months only ~10% of the time (~3 days). Thus annual averages computed directly from the 6h fluxes are more heavily weighted by the periods with high proportion of valid flux measurements. In contrast, annual averages computed from the monthly means give more equal weighting to all the months.

*P8, l 212: give details on how the total CH4 flux was calculated*

This is computed as follows: 0.047 mmol/m2/d * 1400 m * 12429000 m *365 d/yr / 1000 * 16 g/mol = 4.8e9 g/yr. We will specify in the revised ms that we are extrapolating the annual mean from the open water sector (0.047 mmol/m2/d).

*P8, l 216: give details to the reader on how the random instrument noise is calculated. Is the instrumental noise mentioned on line 219 the same noise?*

Yes we are referring to flux uncertainty due to random instrumental noise on both of these occasions. As explained by Yang et al. 2016b, the random uncertainty in the fluxes can be estimated either theoretically (based on instrument's band-limited noise) or using experimental data (by offsetting w and gas mixing ratio, or by choosing a period of zero flux). We will refer interested readers to the detailed analyses in Yang et al. 2016b rather than repeating something similar here.

*P8, l 229: only daytime measurements of pCO2 are mentioned, no night time measurements performed, right?*

Correct.

*P8, l 231: Not clear which data were interpolated, and how they were interpolated*

We will state that pCO2 was measured on the 7th and 12th of July and these data were linearly interpolated to the times of the flux measurements.

*P8, l 235: see comment on page 5. How far further upwind?*

The Kljun et al. (2004) model predicts that the Xmax under these conditions will be greater compared to the neutral condition by a few tens of percent or less.

*P9, l 248-29: again, not clear why mean calculated from monthly mean and from 6h mean are (so) different*

See earlier response.

*P9, l 259: there are many speed-dependant transfer velocity relationships. Choice of the only one from Nightingale et al 2000 paper should be justified. On which basis this choice was made.*

There are obviously numerous gas transfer parameterizations (K) in the literature. We use Nightingale et al. (2000) here not necessarily because it is 'right', but because it is commonly used and lies between the very strong and the very weak wind speed dependent relationships. In section 5 we compute the actual gas transfer velocity from the fluxes and dpCO2, which is then compared against Nightingale et al. (2000) and a few other K parameterizations.

*P9, l 261: wind speed threshold above 5 ms-1 seems quite high. Any justification of that wind speed value?*

A threshold of 5 m/s is used here in the calculation of deltaC = flux / K because at low wind speeds, both the flux and K trend towards zero. Dividing one by the other then obviously leads to large uncertainties.

This is shown in the figure below of the implied CH4 air-sea concentration difference. A 5 m/s wind speed cutoff appears to remove the most obvious ouliers.

[Figure]

*P9 l 264-268: saturation level relative to atmospheric saturation are defined but not used on figure 6. This could be done for the reader to better follow the discussion*

Atmospheric CH4 mixing ratio varies by approximately 10% on a seasonal basis, while CH4 solubility varies by about 14%. As a result of this fairly weak seasonality, a plot of saturation (=Cw/(Ca*H)) looks very similar to a plot of Cw, just with different scales.

[Figure]

We can add a figure of the implied CH4 saturation level in the supplementary material.

*P9, l 268: Is the effect of salinity and temperature accounted for in the 14% variation of CH4 solubility? Not evident on figure 6.*

Yes temperature and salinity are accounted for in the CH4 solubility and equilibrium value calculation. You can see the latter a bit more clearly in the log scale version of figure 6 below.

[Figure]

*P10, l 277: which time series is commented here, 6h or 1h mean data?*

We assume that the reviewer is asking about the semi-diurnal variability in CH4 flux? This is more obviously seen in the 6-hr CH4 flux data, with peak flux values at around 1800 on 9[th] Jul and 0600 on 10[th] Jul (12 hours apart). More generally, adjacent 6-hr CH4 fluxes during this period always alternate between higher values and lower values every six hours.

*P10, l278: same pattern that what? Semi-diurnal variability? That is not possible, this must be something else. . .*

Yes we mean to say that CH4 flux varies on a tidal scale as a result of the outflow from the Tamar estuary. CH4 flux (and saturation) tends to be higher during rising tide than during falling tide.

*P10, l 285-287: comparison is made on two set of data without the same number of monthly data. Not sure it is meaningful.*

We will specify that this comparison (i.e. 32 uatm difference in the mean) is made only during months when we had flux measurements from both sectors.

*P11, l 299-300: seems that the sentence should be reworded*

We have revised the sentence to "It is worth noting that our implied seawater GHG concentrations would be overestimated if the actual gas transfer velocity were higher than the wind speed dependent parameterization of Nightingale et al. (2000)." We are calculating deltaC = flux / K. If the K values we're using is too low (e.g. because it doesn't include contribution from bottom-driven turbulence), we would be getting a deltaC value that is higher than the actual deltaC.

*P11, 309: syntax? Missing word?*

Yes it should've been "we first evaluate the spatial homogeneity of our study region using the shipboard seawater measurements"

*P12, l 350: not the highest saturation, but highest absolute concentration.*

We will specify "the highest dissolved CH4 concentration"

*P12, l 353-354: data at L4 not on figure, could be added for comparison.*

The April and July 2017 L4 data on shown in Figure 11 already.

*P12, l 357: four "times" higher. Times is missing*

Thanks for spotting our error.

*P13, l 359: mentioned that it is the flux footprints for the open water sector*

Suggestion accepted.

*P13, l 383: how did you choose to discard interpolation more than four days away?*

We chose this number because the pCO2 measurements were made approximately once a week. Thus per our filter, we keep the computed kCO2 values if they are sandwiched by pCO2 measurements that were made by a maximum of four days (~half a week) away.

*P14 l 393: how this threshold of 20 µatm was chosen/determined? Any justification?*

A dpCO2 threshold of 40 uatm is commonly used in gas exchange studies.  Though we find that a 20 uatm threshold already filters out most of the noise as a result of dividing by a dpCO2 that is too close to zero. See plot below.

[Figure]

*P14, l 397: be more explicit on what you call "measurements uncertainties in flux, variability in pCO2, as well as processes other than wind speed . . ."*

Measurement uncertainties in flux here are due to random error in the flux measurements.  pCO2 was only measured on approximately a weekly basis, and thus any variability in pCO2 at higher frequencies would contribute to scatter in kCO2.  We will specify that processes other than wind speed are discussed in more detail in the rest of Section 5.

*P14, l 413-414: why did not you check the effect of planar fit vs double rotation to confirm your assumption?*

We did check the effect of planar fit vs double rotation in 2014 when the mast height was higher (27 m above mean sea level).  As shown in the plot below, the u* values from the two methods agree pretty well (ratio around 1.0) at wind speeds over ~5 m/s.  Below this threshold there is a bit more divergence. Because the difference in fluxes doesn't seem to be large between the two methods, and because the double rotation method is commonly used in the air-see exchange community, it is what has been used here as well.  The planar fit angles will be different during this measurement campaign (2015 – 2016) compared to 2014 because the mast height was lower (18 m above mean sea level).  We have not repeated this analysis for the new mast height but aim to do so in the future.

[Figure]

*P15, l 441: what is the effect of the 40µatm threshold on that gap?*

Not sure what the reviewer means here. A threshold on dpCO2 for the kCO2 calculation has no bearing on the actual gaps (i.e. missing data) in the pCO2 observations.

*P15 : the highest Chla measurements in Plymouth Sound is well above the regression line. Any clue for that?*

By regression line, I assume the reviewer is referring to Figure S8? Figure S8 is made when there were near-concurrent measurements of Chla at the two locations. There are occasions when the Chla within the Plymouth Sound were quite different from that at L4 due to natural spatial variability.

*P14, l 467: Be more clear about "to reduce the temporal mismatch between the flux and pCO2 measurements".*

We will remove this unnecessary phrase. The first part of this sentence explains already that since there is a strong diel cycle in the CO2 flux for the Plymouth Sound sector, it makes sense to look only at kCO2 during hours when pCO2 was measured.

*P15, l 479: ". . .could result in biased annual mean flux estimates". Could this be more precisely quantified for example in the case of only daytime measurements?*

This is already presented in the paragraph beginning on line 248 in the case of diurnal variability in CO2 flux. We will refer to Sections 3.1 and 3.2 here.

*Figures P23: use consistent legend throughout the paper with: SW (open water) and NE (Plymouth Sound) all along.*

Suggestion accepted.

*P25: figure 5A: you should not display negative value for K which have no physical meaning. You could shade the night time period on the figure.*

Suggestion accepted.

*P27: figure 9: add units to the X and Y axis*

Suggestion accepted.  We will label them as PSU.

---

## Author Comment (AC2) · 14 Feb 2019

Author Comment with regard to:

"Insights from year-long measurements of air-water CH4 and CO2 exchange in a coastal environment"

by Yang et al.

Many thanks for the detailed *comments and suggestions from Anonymous Referee #2*. Below are our replies to the referee's comments, which are in *italic*.

***Anonymous Referee #2***
*General Comments*
*Yang et al. present an annual monitoring of CH4 and CO2 fluxes from a coastal location in the southwest UK, using the eddy covariance technique. The paper is generally well-written and easy to follow. The manuscript shows important findings related to the investigation of carbon dynamics in coastal zones, which remains largely unknown due to its intrinsic high spatio-temporal variability. The authors showed high differences between fluxes measured at daytime and nighttime, implying that results covering only daytime may be biased due to the influence of biological activities (called diurnal) and tidal processes (called semi-diurnal). The methodology of in situ measurements and data processing are consistent. However, the authors must to address the important appointments from Erik Nilsson and the other anonymous referee. The authors also must improve the description of state of art of CO2 and CH4 dynamics in coastal zones and estuaries (introduction). This section is poorly described. The same appointment is also true for the discussion section related to the dynamics of these GHGs. This part of the manuscript is also unsatisfactorily constructed. The manuscript is well described/structured in terms of technical aspects, but the data interpretation is not sufficient discussed in terms of ecological/biogeochemical processes. A more detailed bibliographical survey is strongly recommended to support your findings and to better contextualize this study.*

*Specific Comments*
*Abstract What it means the semi-diurnal timescales (tidal processes)? Please, describe what is the pCO2 (partial pressure of CO2 in water/air).*

We will specify that 'semi-diurnal' refers to tide-driven variability. We will define pCO2.

*Introduction*
*See the general comments.*
*Lines 31-32: ". . .have been increasing over the last few hundred years primarily due to human activities (Hartmann et al. 2013)." I agree. However, the fastest increase is related to the last 50 years. . .*

We will mention that the fastest increases have occurred over the last 50 years.

*Lines 38-43: This paragraph is a simplistic exposition of the CH4 dynamics/cycling. You should go deep in this part, especially in the studies covering the coastal oceans/estuaries. I recommend a better literature research.*

We will add the following:
CH4 concentrations in estuaries can be influenced by processes including biological productivity, organic carbon input, benthic and particle-derived CH4 production, oxygen content, as well as the hydrodynamics (e.g. Upstill-Goddard et al. 2000; 2016). In regions of intense benthic methanogenesis, gas bubbles supersaturated with CH4 episodically rise through the water column to the surface (e.g. Dimitrov, 2002; Kitidis et al., 2007). This process of ebullition will result in CH4 emissions that are not quantified using air-sea flux calculations based on seawater CH4 concentration. In coastal seas, CH4 saturation tends to be lower than in estuaries, but is still much greater than 100% (e.g. mean >200% for European shelf waters; Bange et al. 2006).

*Lines 45-47: "Globally averaged, the open ocean is modelled to absorb about a quarter of the anthropogenic CO2 emission (Le Quéré et al. 2015)." You can give numbers, and update this reference (from the global carbon budget 2018; Le Quéré et al. 2018). https://www.earth-syst-sci-data.net/10/2141/2018/*

Suggestion accepted.

*Lines 50-56: "Estuaries, on the other hand, are generally net sources of CO2 for the atmosphere (e.g. Frankignoulle et al. 1998). . ." How much? Frankinoulle et al. 1998 is a classical reference. However, there are more recent references that you should include when is describing the global emissions of CO2 by estuaries. In addition, you should shortly describe processes that can affects the dynamics and fluxes at the air- water interface.*

Processes that affect gas exchange at the air-water interface (in terms of delta C and K) are already described between lines 61 and 74. However, we will add the following text:
"Inner estuaries are estimated to emit about 0.3 Pg C yr-1 of CO2 globally (Laruelle et al. 2010; Cai 2011). Most of this CO2 emission is due to the degradation of allochthonous organic matter rather than a direct input of dissolved inorganic carbon (Borges et al. 2006)."

*Lines 55-56: "The shallow seas are predicted to become a greater net sink of CO2 in the future due to rising atmospheric CO2 and increasing inorganic nutrients (e.g. Andersson and Mackenzie, 2004)." However, other studies showed that the estuaries could emit more CO2 due to the enhancing of organic matter respiration.*

We will rephrase to "The coastal seas may have been heterotrophic during preindustrial conditions and thus a net source of CO2 due to organic carbon degradation (e.g. Smith and Hollibaugh, 1993).  Some studies (e.g. Andersson and Mackenzie, 2004; Cai, 2011) predict that shallow seas will become a net sink (or a reduced source) of CO2 in the future due to rising atmospheric CO2 levels and increased inorganic nutrient inputs."

*Lines 73-74: "Thus a wind speed-only dependent representation of K, incomplete for the open ocean (Wanninkhof et al. 2009)." Please rephrase.*

We will rephrase to: "Thus a wind speed-only dependent representation of K is probably less appropriate for coastal environments than for the open ocean."

*Experimental Lines 105-107: "For this paper, wind data from the Windmaster Pro sonic anemometer were used between September 2015 and March 2016. Since March 2016, wind data from the R3 sonic anemometer (not operational for the first 6 months of this annual study) were preferred because of its higher precision and better performance during heavy rain events. Did you compare the wind velocity from the 2 different anemometers used in this study?*

Yes, the wind velocities from two sonic anemometers agree to within 2%.  We will add this information to the supplementary.

*Results Fig. 1. Could you better explain the causes of the negative fluxes of CH4? This means that the water was sub-saturated with respect to atmospheric CH4 concentrations? What are the main causes of this under saturation?*

As shown in Fig. 2, in January 2016 the mean CH4 flux was about zero and the 25th percentile of CH4 flux was about -0.025 mmole/m2/d.  The precision of the CH4 flux was at best 0.01 mmole/m2/d, and tends to worsen at lower wind speeds.  Thus most of what appears to be 'negative' CH4 flux is within the precision of the eddy covariance flux measurement and is not statistically different from zero.

*Could you add a plot combining the results of chl a concentrations and the pCO2 fluxes/values (scatter plot)?*

The plot below shows that pCO2 broadly decreases at higher Chla concentrations. We will add this to the supplementary. There isn't any significant relationship between CO2 flux and Chla, since the CO2 flux is primarily determined by the air-sea pCO2 difference as well as near surface turbulence.

[Figure]

*Lines 210-213: "If the PPAO open water footprint is representative of the nearest 1.4 km (i.e. X90 of our fluxes, see Section 2.2) of the UK coast, our measurements extrapolate to a total CH4 flux of 4.8 Gg yr-1 in UK coastal seas." The extrapolation of the results to other areas is a good exercise. However, CH4 is a gas that present very special conditions of production and consumption. I am not convinced about this calculation.*

We will state that this is only an order-of-magnitude calculation.

*Lines 219-221: "Ambient variability (in e.g. dissolved concentrations) largely drives the rapid temporal fluctuations in CO2 flux, which is unlikely to be fully captured by weekly or monthly seawater sampling." This passage is confuse. Please rewrite.*

We will rephrase to "The rapid temporal fluctuations in CO2 flux are likely to be driven by variability in winds as well as variability in seawater pCO2. The latter is unlikely to be fully captured by weekly or monthly seawater sampling."

*Fig. 4. CO2 flux from the Plymouth Sound sector appeared to be more positive than from the open water sector in some months. I would expect larger differences, but this is not the case. Could you explain this?*

As shown in Fig. 10, pCO2 in the Plymouth Sound sector wasn't very different from pCO2 at L4. It seems that the direct influence of the river Tamar on pCO2 during this period was rather limited.

*Lines 239-242: "Wind speed was generally higher at night during these few days and the measured fluxes imply that ΔpCO2 (see next sections on this calculation) changed from about -40 µatm during the day to about 15 µatm at night." Then, during this period, the system was a net sink of CO2?*

The mean CO2 flux during these three days was small (about -2 mmol/m2/d).

*Lines 283-285: "The greatest undersaturation in CO2 is observed in late spring and early summer, coinciding with an increase in chlorophyll a concentration at the nearby L4 station (Figure S6)." As I said before, I would like to see graph showing the chl a and the CO2 concentrations.*

Please see above.

*Lines 295-298: "The diurnal variability we observed is important in the context of estuarine CO2 (and carbonate system) observations that are predominantly carried out during daytime when sampling and navigation are easier." Many studies were published recently covering the diurnal (biological effect) and the semi-diurnal (tidal effect) variability on pCO2 changes, which are poorly described and constrained in the present manuscript.*

We will rephrase this sentence to "The diurnal variability we observed is important in the context of the estuarine CO2 (and carbonate system) observations that are only made during the daytime." We will also add references of some recent studies that look at the diurnal and tidal-driven variability in pCO2 where appropriate.

*Line 304: Spatial variability in seawater concentrations. Please see the general comments. This could be because the influence of Tamar estuary on the PPAO flux footprint is less in terms of pCO2 (e.g. due to the already large burden of carbonate and bicarbonate in seawater), and more on physical and biogeochemical processes. I am not sure about this statement.*

We will rephrase this sentence to "pCO$_2$ measurements within both flux footprints were very similar to pCO$_2$ at the L4 station.", and remove the latter phrase.

*Line 306: "dissolved pCO2." This is unusual. You should refer "pCO2" or "dissolved CO2".*

We will rephrase to seawater pCO2.

*Lines 329-331: "It could be that the large burden of carbonate and bicarbonate in seawater partially buffered the impact of the freshwater input on pCO2 within the flux footprints." This is poorly discussed.*

We will remove this sentence as a similar point is already made at the end of section 3.4.

*Lines 373- 377 "The implied pCO2 from EC fluxes in monthly bins and in situ measured pCO2 agree quite well over the annual cycle for the open water sector (Figure 7), suggesting that the use of the wind speed dependent transfer velocity parameterization of Nightingale et al. (2000) is largely reasonable." I am not sure that this type o graph is the best to show a comparison between estimated and measured pCO2.*

We think this is a reasonable way to show both the seasonality in pCO2 and qualitatively compare the implied (=flux/K_Nightingale) against the measured pCO2.  A more quantitative comparison between the implied and measured pCO2 is in essence made in the calculation of kCO2,660 (which is then compared against the Nightingale et al. 2000 K parameterization).

*Figure 7. Y-axis : seawater pCO2 ?*

Agree changing to seawater pCO2.

---

## Author Response (AR1)

Dear editor:

Thank you for your efforts.

The track change version of the revised manuscript is copied below.

Sincerely,

Mingxi Yang et al.

[revised manuscript text omitted]

Mingxi Yang 1/24/19 4:44 PM

---

## Author Response (AR2)

**Response to editors suggested changes:**

*You have chosen the "result and discussion" structure for your paper*
*Chapter numbers and title must follow this format.*
*In addition, I believe your paper be easier to read if you combine actual 3.1+3.3 and 3.2+3.4, without altering the text.*
This is a good suggestion and we have revised the structure of the paper as follows:

> 1 Introduction
> 2 Experimental
> 2.1 Eddy covariance fluxes
> 2.2 Flux footprints
> 2.3 Seawater measurements
> 3 Results and discussion
> 3.1 CH4 fluxes and implied seawater concentrations
> 3.2 CO2 fluxes and implied seawater concentrations
> 3.3 Spatial homogeneity of the study region
> 3.3.1 Variability in salinity
> 3.3.2 Variability in seawater pCO2
> 3.3.3 Variability in dissolved CH4
> 3.4 CO2 gas transfer velocity
> 3.4.1 Dependence of KCO2,660  on wind speed and friction velocity
> 3.4.2 Seasonal variability in KCO2,660
> 3.4.3 Dependence of KCO2,660  on bottom-driven turbulence
> 3.5 Effects of rain on air-sea CO2 exchange
> 4 Conclusions

*L39 : change "terrestrial aquatic systems" to "continental aquatic systems" or "inland aquatic systems"*
Changed to "Inland aquatic systems"

*L50 "Due to both the 'solubility pump' and the 'biological pump', the surface ocean can be a net source or sink of CO2, depending on location and time of the year" The term "pump" clearly applies for the case of a sink of atmospheric CO2, but not for a source. Please rephrase*
Revised to:
"Seawater $CO_2$ levels are primarily determined by solubility (temperature-dependent) and the balance between primary production and respiration by the biological community.  Seasonal and geographical differences in seawater temperature and biological activity mean that the surface ocean can act as a net source or sink of $CO_2$, depending on location and time of the year (Khatiwala et al. 2013; Houghton 2003)."

*L52"The global open ocean is modelled to absorb", not sure this is the appropriate formulation*
Revised to:
"Models estimate that 2.4±0.5 GtC $yr^{-1}$ of $CO_2$ (a quarter of anthropogenic emissions) have been absorbed by the global ocean over the last decade (Le Quéré et al. 2018).

*L83 not sure if mentioning such details on surfactants effect on K is relevant in the introduction of the present study, which does not deal with surfactants.*

The section on seasonal variability in $K_{CO2,660}$ (now Section 3.4.2) discusses this briefly and refers to Figure 13 ($K_{CO2,660}$ colour-coded by Chl *a*, which is often used as an indicator of likely surfactant levels). We feel that this is a relevant-enough topic to be retained in the introduction.